# Breaking the Nonsmooth Barrier: A Scalable Parallel Method for Composite Optimization

**Fabian Pedregosa**
INRIA/ENS[*]
Paris, France

**Rémi Leblond**
INRIA/ENS[*]
Paris, France

**Simon Lacoste-Julien**
MILA and DIRO
Université de Montréal, Canada

## Abstract

Due to their simplicity and excellent performance, parallel asynchronous variants of stochastic gradient descent have become popular methods to solve a wide range of large-scale optimization problems on multi-core architectures. Yet, despite their practical success, support for nonsmooth objectives is still lacking, making them unsuitable for many problems of interest in machine learning, such as the Lasso, group Lasso or empirical risk minimization with convex constraints. In this work, we propose and analyze PROXASAGA, a fully asynchronous sparse method inspired by SAGA, a variance reduced incremental gradient algorithm. The proposed method is easy to implement and significantly outperforms the state of the art on several nonsmooth, large-scale problems. We prove that our method achieves a theoretical linear speedup with respect to the sequential version under assumptions on the sparsity of gradients and block-separability of the proximal term. Empirical benchmarks on a multi-core architecture illustrate practical speedups of up to 12x on a 20-core machine.

## 1 Introduction

The widespread availability of multi-core computers motivates the development of parallel methods adapted for these architectures. One of the most popular approaches is HOGWILD (Niu et al., 2011), an asynchronous variant of stochastic gradient descent (SGD). In this algorithm, multiple threads run the update rule of SGD asynchronously in parallel. As SGD, it only requires visiting a small batch of random examples per iteration, which makes it ideally suited for large scale machine learning problems. Due to its simplicity and excellent performance, this parallelization approach has recently been extended to other variants of SGD with better convergence properties, such as SVRG (Johnson & Zhang, 2013) and SAGA (Defazio et al., 2014).

Despite their practical success, existing parallel asynchronous variants of SGD are limited to smooth objectives, making them inapplicable to many problems in machine learning and signal processing. In this work, we develop a sparse variant of the SAGA algorithm and consider its parallel asynchronous variants for general *composite* optimization problems of the form:

$$\arg\min_{\boldsymbol{x}\in\mathbb{R}^p} f(\boldsymbol{x}) + h(\boldsymbol{x}) \quad , \quad \text{with } f(\boldsymbol{x}) := \tfrac{1}{n}\sum_{i=1}^n f_i(\boldsymbol{x}) \quad , \qquad \text{(OPT)}$$

where each $f_i$ is convex with $L$-Lipschitz gradient, the average function $f$ is $\mu$-strongly convex and $h$ is convex but potentially nonsmooth. We further assume that $h$ is "simple" in the sense that we have access to its proximal operator, and that it is block-separable, that is, it can be decomposed block coordinate-wise as $h(\boldsymbol{x}) = \sum_{B\in\mathcal{B}} h_B([\boldsymbol{x}]_B)$, where $\mathcal{B}$ is a partition of the coefficients into

---

[*]DI École normale supérieure, CNRS, PSL Research University

subsets which will call *blocks* and $h_B$ only depends on coordinates in block $B$. Note that there is no loss of generality in this last assumption as a unique block covering all coordinates is a valid partition, though in this case, our sparse variant of the SAGA algorithm reduces to the original SAGA algorithm and no gain from sparsity is obtained.

This template models a broad range of problems arising in machine learning and signal processing: the finite-sum structure of $f$ includes the least squares or logistic loss functions; the proximal term $h$ includes penalties such as the $\ell_1$ or group lasso penalty. Furthermore, this term can be extended-valued, thus allowing for convex constraints through the indicator function.

**Contributions.** This work presents two main contributions. First, in §2 we describe Sparse Proximal SAGA, a novel variant of the SAGA algorithm which features a reduced cost per iteration in the presence of sparse gradients and a block-separable penalty. Like other variance reduced methods, it enjoys a linear convergence rate under strong convexity. Second, in §3 we present PROXASAGA, a lock-free asynchronous parallel version of the aforementioned algorithm that does not require consistent reads. Our main results states that PROXASAGA obtains (under assumptions) a theoretical linear speedup with respect to its sequential version. Empirical benchmarks reported in §4 show that this method dramatically outperforms state-of-the-art alternatives on large sparse datasets, while the empirical speedup analysis illustrates the practical gains as well as its limitations.

## 1.1 Related work

**Asynchronous coordinate-descent.** For composite objective functions of the form (OPT), most of the existing literature on asynchronous optimization has focused on variants of coordinate descent. Liu & Wright (2015) proposed an asynchronous variant of (proximal) coordinate descent and proved a near-linear speedup in the number of cores used, given a suitable step size. This approach has been recently extended to general block-coordinate schemes by Peng et al. (2016), to greedy coordinate-descent schemes by You et al. (2016) and to non-convex problems by Davis et al. (2016). However, as illustrated by our experiments, in the large sample regime coordinate descent compares poorly against incremental gradient methods like SAGA.

**Variance reduced incremental gradient and their asynchronous variants.** Initially proposed in the context of smooth optimization by Le Roux et al. (2012), variance reduced incremental gradient methods have since been extended to minimize composite problems of the form (OPT) (see table below). Smooth variants of these methods have also recently been extended to the asynchronous setting, where multiple threads run the update rule asynchronously and in parallel. Interestingly, none of these methods achieve both simultaneously, i.e. asynchronous optimization of composite problems. Since variance reduced incremental gradient methods have shown state of the art performance in both settings, this generalization is of key practical interest.

| Objective | Sequential Algorithm | Asynchronous Algorithm |
|---|---|---|
| Smooth | SVRG (Johnson & Zhang, 2013) | SVRG (Reddi et al., 2015) |
| | SDCA (Shalev-Shwartz & Zhang, 2013) | PASSCODE (Hsieh et al., 2015, SDCA variant) |
| | SAGA (Defazio et al., 2014) | ASAGA (Leblond et al., 2017, SAGA variant) |
| Composite | PROXSDCA (Shalev-Shwartz et al., 2012) | |
| | SAGA (Defazio et al., 2014) | This work: PROXASAGA |
| | PROXSVRG (Xiao & Zhang, 2014) | |

**On the difficulty of a composite extension.** Two key issues explain the paucity in the development of asynchronous incremental gradient methods for composite optimization. The first issue is related to the design of such algorithms. Asynchronous variants of SGD are most competitive when the updates are sparse and have a small overlap, that is, when each update modifies a small and different subset of the coefficients. This is typically achieved by updating only coefficients for which the partial gradient at a given iteration is nonzero,[2] but existing schemes such as the lagged updates technique (Schmidt et al., 2016) are not applicable in the asynchronous setting. The second

difficulty is related to the analysis of such algorithms. All convergence proofs crucially use the Lipschitz condition on the gradient to bound the noise terms derived from asynchrony. However, in the composite case, the gradient mapping term (Beck & Teboulle, 2009), which replaces the gradient in proximal-gradient methods, does not have a bounded Lipschitz constant. Hence, the traditional proof technique breaks down in this scenario.

**Other approaches.** Recently, Meng et al. (2017); Gu et al. (2016) independently proposed a doubly stochastic method to solve the problem at hand. Following Meng et al. (2017) we refer to it as Async-PROXSVRCD. This method performs coordinate descent-like updates in which the true gradient is replaced by its SVRG approximation. It hence features a doubly-stochastic loop: at each iteration we select a random coordinate *and* a random sample. Because the selected coordinate block is uncorrelated with the chosen sample, the algorithm can be orders of magnitude slower than SAGA in the presence of sparse gradients. Appendix F contains a comparison of these methods.

## 1.2 Definitions and notations

By convention, we denote vectors and vector-valued functions in lowercase boldface (e.g. $\boldsymbol{x}$) and matrices in uppercase boldface (e.g. $\boldsymbol{D}$). The proximal operator of a convex lower semicontinuous function $h$ is defined as $\mathbf{prox}_h(\boldsymbol{x}) := \arg\min_{\boldsymbol{z} \in \mathbb{R}^p} \{h(\boldsymbol{z}) + \frac{1}{2}\|\boldsymbol{x} - \boldsymbol{z}\|^2\}$. A function $f$ is said to be $L$-smooth if it is differentiable and its gradient is $L$-Lipschitz continuous. A function $f$ is said to be $\mu$-strongly convex if $f - \frac{\mu}{2}\|\cdot\|^2$ is convex. We use the notation $\kappa := L/\mu$ to denote the condition number for an $L$-smooth and $\mu$-strongly convex function.[3]

$\boldsymbol{I}_p$ denotes the $p$-dimensional identity matrix, $\mathbb{1}\{\text{cond}\}$ the characteristic function, which is 1 if cond evaluates to true and 0 otherwise. The average of a vector or matrix is denoted $\overline{\boldsymbol{\alpha}} := \frac{1}{n}\sum_{i=1}^n \boldsymbol{\alpha}_i$. We use $\|\cdot\|$ for the Euclidean norm. For a positive semi-definite matrix $\boldsymbol{D}$, we define its associated distance as $\|\boldsymbol{x}\|_{\boldsymbol{D}}^2 := \langle \boldsymbol{x}, \boldsymbol{D}\boldsymbol{x}\rangle$. We denote by $[\boldsymbol{x}]_b$ the $b$-th coordinate in $\boldsymbol{x}$. This notation is overloaded so that for a collection of blocks $T = \{B_1, B_2, \ldots\}$, $[\boldsymbol{x}]_T$ denotes the vector $\boldsymbol{x}$ restricted to the coordinates in the blocks of $T$. For convenience, when $T$ consists of a single block $B$ we use $[\boldsymbol{x}]_B$ as a shortcut of $[\boldsymbol{x}]_{\{B\}}$. Finally, we distinguish $\mathbb{E}$, the full expectation taken with respect to all the randomness in the system, from $\mathbf{E}$, the conditional expectation of a random $i_t$ (the random index sampled at each iteration by SGD-like algorithms) conditioned on all the "past", which the context will clarify.

## 2 Sparse Proximal SAGA

**Original SAGA algorithm.** The original SAGA algorithm (Defazio et al., 2014) maintains two moving quantities: the current iterate $\boldsymbol{x}$ and a table (memory) of historical gradients $(\boldsymbol{\alpha}_i)_{i=1}^n$. At every iteration, it samples an index $i \in \{1, \ldots, n\}$ uniformly at random, and computes the next iterate $(\boldsymbol{x}^+, \boldsymbol{\alpha}^+)$ according to the following recursion:

$$\boldsymbol{u}_i = \nabla f_i(\boldsymbol{x}) - \boldsymbol{\alpha}_i + \overline{\boldsymbol{\alpha}}; \quad \boldsymbol{x}^+ = \mathbf{prox}_{\gamma h}(\boldsymbol{x} - \gamma \boldsymbol{u}_i); \quad \boldsymbol{\alpha}_i^+ = \nabla f_i(\boldsymbol{x}). \tag{1}$$

On each iteration, this update rule requires to visit all coefficients even if the partial gradients $\nabla f_i$ are sparse. Sparse partial gradients arise in a variety of practical scenarios: for example, in generalized linear models the partial gradients inherit the sparsity pattern of the dataset. Given that large-scale datasets are often sparse,[4] leveraging this sparsity is crucial for the success of the optimizer.

**Sparse Proximal SAGA algorithm.** We will now describe an algorithm that leverages sparsity in the partial gradients by only updating those blocks that intersect with the support of the partial gradients. Since in this update scheme some blocks might appear more frequently than others, we will need to counterbalance this undersirable effect with a well-chosen block-wise reweighting of the average gradient and the proximal term.

In order to make precise this block-wise reweighting, we define the following quantities. We denote by $T_i$ the *extended support* of $\nabla f_i$, which is the set of blocks that intersect the support of $\nabla f_i$,

formally defined as $T_i := \{B : \mathrm{supp}(\nabla f_i) \cap B \neq \varnothing,\, B \in \mathcal{B}\}$. For totally separable penalties such as the $\ell_1$ norm, the blocks are individual coordinates and so the extended support covers the same coordinates as the support. Let $d_B := n/n_B$, where $n_B := \sum_i \mathbb{1}\{B \in T_i\}$ is the number of times that $B \in T_i$. For simplicity we assume $n_B > 0$, as otherwise the problem can be reformulated without block $B$. The update rule in (1) requires computing the proximal operator of $h$, which involves a full pass on the coordinates. In our proposed algorithm, we replace $h$ in (1) with the function $\varphi_i(x) := \sum_{B \in T_i} d_B h_B(x)$, whose form is justified by the following three properties. First, this function is zero outside $T_i$, allowing for sparse updates. Second, because of the block-wise reweighting $d_B$, the function $\varphi_i$ is an unbiased estimator of $h$ (i.e., $\mathbf{E}\,\varphi_i = h$), property which will be crucial to prove the convergence of the method. Third, $\varphi_i$ inherits the block-wise structure of $h$ and its proximal operator can be computed from that of $h$ as $[\mathbf{prox}_{\gamma\varphi_i}(x)]_B = [\mathbf{prox}_{(d_B\gamma)h_B}(x)]_B$ if $B \in T_i$ and $[\mathbf{prox}_{\gamma\varphi_i}(x)]_B = [x]_B$ otherwise. Following Leblond et al. (2017), we will also replace the dense gradient estimate $u_i$ by the sparse estimate $v_i := \nabla f_i(x) - \alpha_i + D_i\overline{\alpha}$, where $D_i$ is the diagonal matrix defined block-wise as $[D_i]_{B,B} = d_B \mathbb{1}\{B \in T_i\} I_{|B|}$. It is easy to verify that the vector $D_i\overline{\alpha}$ is a weighted projection onto the support of $T_i$ and $\mathbf{E}\,D_i\overline{\alpha} = \overline{\alpha}$, making $v_i$ an unbiased estimate of the gradient.

We now have all necessary elements to describe the Sparse Proximal SAGA algorithm. As the original SAGA algorithm, it maintains two moving quantities: the current iterate $x \in \mathbb{R}^p$ and a table of historical gradients $(\alpha_i)_{i=1}^n$, $\alpha_i \in \mathbb{R}^p$. At each iteration, the algorithm samples an index $i \in \{1, \ldots, n\}$ and computes the next iterate $(x^+, \alpha^+)$ as:

$$v_i = \nabla f_i(x) - \alpha_i + D_i\overline{\alpha}\,;\; x^+ = \mathbf{prox}_{\gamma\varphi_i}\big(x - \gamma v_i\big)\,;\; \alpha_i^+ = \nabla f_i(x)\,, \qquad \text{(SPS)}$$

where in a practical implementation the vector $\overline{\alpha}$ is updated incrementally at each iteration.

The above algorithm is sparse in the sense that it only requires to visit and update blocks in the extended support: if $B \notin T_i$, by the sparsity of $v_i$ and $\mathbf{prox}_{\varphi_i}$, we have $[x^+]_B = [x]_B$. Hence, when the extended support $T_i$ is sparse, this algorithm can be orders of magnitude faster than the naive SAGA algorithm. The extended support is sparse for example when the partial gradients are sparse and the penalty is separable, as is the case of the $\ell_1$ norm or the indicator function over a hypercube, or when the the penalty is block-separable in a way such that only a small subset of the blocks overlap with the support of the partial gradients. Initialization of variables and a reduced storage scheme for the memory are discussed in the implementation details section of Appendix E.

**Relationship with existing methods**. This algorithm can be seen as a generalization of both the Standard SAGA algorithm and the Sparse SAGA algorithm of Leblond et al. (2017). When the proximal term is not block-separable, then $d_B = 1$ (for a unique block $B$) and the algorithm defaults to the Standard (dense) SAGA algorithm. In the smooth case (i.e., $h = 0$), the algorithm defaults to the Sparse SAGA method. Hence we note that the sparse gradient estimate $v_i$ in our algorithm is the same as the one proposed in Leblond et al. (2017). However, we emphasize that a straightforward combination of this sparse update rule with the proximal update from the Standard SAGA algorithm results in a nonconvergent algorithm: the block-wise reweighting of $h$ is a surprisingly simple but crucial change. We now give the convergence guarantees for this algorithm.

**Theorem 1.** *Let $\gamma = \frac{a}{5L}$ for any $a \leq 1$ and $f$ be $\mu$-strongly convex ($\mu > 0$). Then Sparse Proximal SAGA converges geometrically in expectation with a rate factor of at least $\rho = \frac{1}{5}\min\{\frac{1}{n}, a\frac{1}{\kappa}\}$. That is, for $x_t$ obtained after $t$ updates, we have the following bound:*

$$\mathbb{E}\|x_t - x^*\|^2 \leq (1 - \rho)^t C_0\,, \quad \text{with } C_0 := \|x_0 - x^*\|^2 + \frac{1}{5L^2}\sum_{i=1}^n \|\alpha_i^0 - \nabla f_i(x^*)\|^2\quad.$$

**Remark**. For the step size $\gamma = 1/5L$, the convergence rate is $(1 - 1/5\min\{1/n, 1/\kappa\})$. We can thus identify two regimes: the "big data" regime, $n \geq \kappa$, in which the rate factor is bounded by $1/5n$, and the "ill-conditioned" regime, $\kappa \geq n$, in which the rate factor is bounded by $1/5\kappa$. This rate roughly matches the rate obtained by Defazio et al. (2014). While the step size bound of $1/5L$ is slightly smaller than the $1/3L$ one obtained in that work, this can be explained by their stronger assumptions: each $f_i$ is strongly convex whereas they are strongly convex only on average in this work. All proofs for this section can be found in Appendix B.

| **Algorithm 1** PROXASAGA (analyzed) | **Algorithm 2** PROXASAGA (implemented) |
|---|---|
| 1: Initialize shared variables $\boldsymbol{x}$ and $(\boldsymbol{\alpha}_i)_{i=1}^n$ | 1: Initialize shared variables $\boldsymbol{x}, (\boldsymbol{\alpha}_i)_{i=1}^n, \overline{\boldsymbol{\alpha}}$ |
| 2: **keep doing in parallel** | 2: **keep doing in parallel** |
| 3:    $\hat{\boldsymbol{x}} =$ inconsistent read of $\boldsymbol{x}$ | 3:    *Sample $i$ uniformly in $\{1, ..., n\}$* |
| 4:    $\hat{\boldsymbol{\alpha}} =$ inconsistent read of $\boldsymbol{\alpha}$ | 4:    $S_i :=$ support of $\nabla f_i$ |
| 5:    *Sample $i$ uniformly in $\{1, ..., n\}$* | 5:    $T_i :=$ extended support of $\nabla f_i$ in $\mathcal{B}$ |
| 6:    $S_i :=$ support of $\nabla f_i$ | 6:    $[\hat{\boldsymbol{x}}]_{T_i} =$ inconsistent read of $\boldsymbol{x}$ on $T_i$ |
| 7:    $T_i :=$ extended support of $\nabla f_i$ in $\mathcal{B}$ | 7:    $\hat{\boldsymbol{\alpha}}_i =$ inconsistent read of $\boldsymbol{\alpha}_i$ |
| 8:    $[\overline{\boldsymbol{\alpha}}]_{T_i} = {}^1\!/n \sum_{j=1}^n [\hat{\boldsymbol{\alpha}}_j]_{T_i}$ | 8:    $[\overline{\boldsymbol{\alpha}}]_{T_i} =$ inconsistent read of $\overline{\boldsymbol{\alpha}}$ on $T_i$ |
| 9:    $[\delta\boldsymbol{\alpha}]_{S_i} = [\nabla f_i(\hat{\boldsymbol{x}})]_{S_i} - [\hat{\boldsymbol{\alpha}}_i]_{S_i}$ | 9:    $[\delta\boldsymbol{\alpha}]_{S_i} = [\nabla f_i(\hat{\boldsymbol{x}})]_{S_i} - [\hat{\boldsymbol{\alpha}}_i]_{S_i}$ |
| 10:   $[\hat{\boldsymbol{v}}]_{T_i} = [\delta\boldsymbol{\alpha}]_{T_i} + [\boldsymbol{D}_i\overline{\boldsymbol{\alpha}}]_{T_i}$ | 10:   $[\hat{\boldsymbol{v}}]_{T_i} = [\delta\boldsymbol{\alpha}]_{T_i} + [\boldsymbol{D}_i\overline{\boldsymbol{\alpha}}]_{T_i}$ |
| 11:   $[\delta\boldsymbol{x}]_{T_i} = [\mathbf{prox}_{\gamma\varphi_i}(\hat{\boldsymbol{x}} - \gamma\hat{\boldsymbol{v}})]_{T_i} - [\hat{\boldsymbol{x}}]_{T_i}$ | 11:   $[\delta\boldsymbol{x}]_{T_i} = [\mathbf{prox}_{\gamma\varphi_i}(\hat{\boldsymbol{x}} - \gamma\hat{\boldsymbol{v}})]_{T_i} - [\hat{\boldsymbol{x}}]_{T_i}$ |
| 12:   **for** $B$ **in** $T_i$ **do** | 12:   **for** $B$ **in** $T_i$ **do** |
| 13:     **for** $b \in B$ **do** | 13:     **for** $b$ **in** $B$ **do** |
| 14:       $[\boldsymbol{x}]_b \leftarrow [\boldsymbol{x}]_b + [\delta\boldsymbol{x}]_b$     ▷ atomic | 14:       $[\boldsymbol{x}]_b \leftarrow [\boldsymbol{x}]_b + [\delta\boldsymbol{x}]_b$     ▷ atomic |
| 15:       **if** $b \in S_i$ **then** | 15:       **if** $b \in S_i$ **then** |
| 16:         $[\boldsymbol{\alpha}_i]_b \leftarrow [\nabla f_i(\hat{\boldsymbol{x}})]_b$ | 16:         $[\overline{\boldsymbol{\alpha}}]_b \leftarrow [\overline{\boldsymbol{\alpha}}]_b + {}^1\!/n[\delta\boldsymbol{\alpha}]_b$   ▷ atomic |
| 17:       **end if** | 17:       **end if** |
| 18:     **end for** | 18:     **end for** |
| 19:   **end for** | 19:   **end for** |
| 20:   // ('$\leftarrow$' denotes shared memory update.) | 20:   $\boldsymbol{\alpha}_i \leftarrow \nabla f_i(\hat{\boldsymbol{x}})$   (scalar update)   ▷ atomic |
| 21: **end parallel loop** | 21: **end parallel loop** |

# 3 Asynchronous Sparse Proximal SAGA

We introduce PROXASAGA – the asynchronous parallel variant of Sparse Proximal SAGA. In this algorithm, multiple cores update a central parameter vector using the Sparse Proximal SAGA introduced in the previous section, and updates are performed asynchronously. The algorithm parameters are read and written without vector locks, i.e., the vector content of the shared memory can potentially change while a core is reading or writing to main memory coordinate by coordinate. These operations are typically called *inconsistent* (at the vector level).

The full algorithm is described in Algorithm 1 for its theoretical version (on which our analysis is built) and in Algorithm 2 for its practical implementation. The practical implementation differs from the analyzed agorithm in three points. First, in the implemented algorithm, index $i$ is sampled before reading the coefficients to minimize memory access since only the extended support needs to be read. Second, since our implementation targets generalized linear models, the memory $\boldsymbol{\alpha}_i$ can be compressed into a single scalar in L20 (see Appendix E). Third, $\overline{\boldsymbol{\alpha}}$ is stored in memory and updated incrementally instead of recomputed at each iteration.

The rest of the section is structured as follows: we start by describing our framework of analysis; we then derive essential properties of PROXASAGA along with a classical delay assumption. Finally, we state our main convergence and speedup result.

## 3.1 Analysis framework

As in most of the recent asynchronous optimization literature, we build on the hardware model introduced by Niu et al. (2011), with multiple cores reading and writing to a shared memory parameter vector. These operations are asynchronous (lock-free) and *inconsistent:*[5] $\hat{\boldsymbol{x}}_t$, the local copy of the parameters of a given core, does not necessarily correspond to a consistent iterate in memory.

**"Perturbed" iterates.** To handle this additional difficulty, contrary to most contributions in this field, we choose the "perturbed iterate framework" proposed by Mania et al. (2017) and refined by Leblond et al. (2017). This framework can analyze variants of SGD which obey the update rule:

$$\boldsymbol{x}_{t+1} = \boldsymbol{x}_t - \gamma\boldsymbol{v}(\boldsymbol{x}_t, i_t), \quad \text{where } \boldsymbol{v} \text{ verifies the unbiasedness condition } \mathbf{E}\,\boldsymbol{v}(\boldsymbol{x}, i_t) = \nabla f(\boldsymbol{x})$$

and the expectation is computed with respect to $i_t$. In the asynchronous parallel setting, cores are reading inconsistent iterates from memory, which we denote $\hat{x}_t$. As these inconsistent iterates are affected by various delays induced by asynchrony, they cannot easily be written as a function of their previous iterates. To alleviate this issue, Mania et al. (2017) choose to introduce an additional quantity for the purpose of the analysis:

$$x_{t+1} := x_t - \gamma v(\hat{x}_t, i_t), \quad \text{the "virtual iterate" -- which is never actually computed}. \quad (2)$$

Note that this equation is the *definition* of this new quantity $x_t$. This virtual iterate is useful for the convergence analysis and makes for much easier proofs than in the related literature.

**"After read" labeling.** How we choose to define the iteration counter $t$ to label an iterate $x_t$ matters in the analysis. In this paper, we follow the "after read" labeling proposed in Leblond et al. (2017), in which we update our iterate counter, $t$, as each core *finishes reading* its copy of the parameters (in the specific case of PROXASAGA, this includes both $\hat{x}_t$ and $\hat{\alpha}^t$). This means that $\hat{x}_t$ is the $(t+1)^{th}$ fully completed read. One key advantage of this approach compared to the classical choice of Niu et al. (2011) -- where $t$ is increasing after each successful update -- is that it guarantees both that the $i_t$ are uniformly distributed and that $i_t$ and $\hat{x}_t$ are independent. This property is not verified when using the "after write" labeling of Niu et al. (2011), although it is still implicitly assumed in the papers using this approach, see Leblond et al. (2017, Section 3.2) for a discussion of issues related to the different labeling schemes.

**Generalization to composite optimization.** Although the perturbed iterate framework was designed for gradient-based updates, we can extend it to proximal methods by remarking that in the sequential setting, proximal stochastic gradient descent and its variants can be characterized by the following similar update rule:

$$x_{t+1} = x_t - \gamma g(x_t, v_{i_t}, i_t), \quad \text{with } g(x, v, i) := \frac{1}{\gamma}\left(x - \mathbf{prox}_{\gamma\varphi_i}(x - \gamma v)\right), \quad (3)$$

where as before $v$ verifies the unbiasedness condition $\mathbf{E}\, v = \nabla f(x)$. The Proximal Sparse SAGA iteration can be easily written within this template by using $\varphi_i$ and $v_i$ as defined in §2. Using this definition of $g$, we can define PROXASAGA virtual iterates as:

$$x_{t+1} := x_t - \gamma g(\hat{x}_t, \hat{v}^t_{i_t}, i_t), \quad \text{with } \hat{v}^t_{i_t} = \nabla f_{i_t}(\hat{x}_t) - \hat{\alpha}^t_{i_t} + D_{i_t}\overline{\alpha}^t \quad, \quad (4)$$

where as in the sequential case, the memory terms are updated as $\hat{\alpha}^t_{i_t} = \nabla f_{i_t}(\hat{x}_t)$. Our theoretical analysis of PROXASAGA will be based on this definition of the virtual iterate $x_{t+1}$.

## 3.2 Properties and assumptions

Now that we have introduced the "after read" labeling for proximal methods in Eq. (4), we can leverage the framework of Leblond et al. (2017, Section 3.3) to derive essential properties for the analysis of PROXASAGA. We describe below three useful properties arising from the definition of Algorithm 1, and then state a central (but standard) assumption that the delays induced by the asynchrony are uniformly bounded.

**Independence:** Due to the "after read" global ordering, $i_r$ is independent of $\hat{x}_t$ for all $r \geq t$. We enforce the independence for $r = t$ by having the cores read all the shared parameters before their iterations.

**Unbiasedness:** The term $\hat{v}^t_{i_t}$ is an unbiased estimator of the gradient of $f$ at $\hat{x}_t$. This property is a consequence of the independence between $i_t$ and $\hat{x}_t$.

**Atomicity:** The shared parameter coordinate update of $[x]_b$ on Line 14 is atomic. This means that there are no overwrites for a single coordinate even if several cores compete for the same resources. Most modern processors have support for atomic operations with minimal overhead.

**Bounded overlap assumption.** We assume that there exists a uniform bound, $\tau$, on the maximum number of overlapping iterations. This means that every coordinate update from iteration $t$ is successfully written to memory before iteration $t + \tau + 1$ starts. Our result will give us conditions on $\tau$ to obtain linear speedups.

**Bounding $\hat{x}_t - x_t$.** The delay assumption of the previous paragraph allows to express the difference between real and virtual iterate using the gradient mapping $g_u := g(\hat{x}_u, \hat{v}^u_{i_u}, i_u)$ as:

$$\hat{x}_t - x_t = \gamma \sum_{u=(t-\tau)_+}^{t-1} G^t_u g_u \,, \text{ where } G^t_u \text{ are } p \times p \text{ diagonal matrices with terms in } \{0, +1\}. \quad (5)$$

0 represents instances where both $\hat{\boldsymbol{x}}_u$ and $\boldsymbol{x}_u$ have received the corresponding updates. $+1$, on the contrary, represents instances where $\hat{\boldsymbol{x}}_u$ has not yet received an update that is already in $\boldsymbol{x}_u$ by definition. This bound will prove essential to our analysis.

### 3.3 Analysis

In this section, we state our convergence and speedup results for PROXASAGA. The full details of the analysis can be found in Appendix C. Following Niu et al. (2011), we introduce a sparsity measure (generalized to the composite setting) that will appear in our results.

**Definition 1.** *Let* $\Delta := \max_{B \in \mathcal{B}} |\{i : T_i \ni B\}|/n$. *This is the normalized maximum number of times that a block appears in the extended support. For example, if a block is present in all* $T_i$, *then* $\Delta = 1$. *If no two* $T_i$ *share the same block, then* $\Delta = 1/n$. *We always have* $1/n \leq \Delta \leq 1$.

**Theorem 2** (Convergence guarantee of PROXASAGA). *Suppose* $\tau \leq \frac{1}{10\sqrt{\Delta}}$. *For any step size* $\gamma = \frac{a}{L}$ *with* $a \leq a^*(\tau) := \frac{1}{36} \min\{1, \frac{6\kappa}{\tau}\}$, *the inconsistent read iterates of Algorithm 1 converge in expectation at a geometric rate factor of at least:* $\rho(a) = \frac{1}{5} \min\left\{\frac{1}{n}, a\frac{1}{\kappa}\right\}$, *i.e.* $\mathbb{E}\|\hat{\boldsymbol{x}}_t - \boldsymbol{x}^*\|^2 \leq (1-\rho)^t \tilde{C}_0$, *where* $\tilde{C}_0$ *is a constant independent of* $t$ $(\approx \frac{n\kappa}{a} C_0$ *with* $C_0$ *as defined in Theorem ??).*

This last result is similar to the original SAGA convergence result and our own Theorem **??**, with both an extra condition on $\tau$ and on the maximum allowable step size. In the best sparsity case, $\Delta = 1/n$ and we get the condition $\tau \leq \sqrt{n}/10$. We now compare the geometric rate above to the one of Sparse Proximal SAGA to derive the necessary conditions under which PROXASAGA is linearly faster.

**Corollary 1** (Speedup). *Suppose* $\tau \leq \frac{1}{10\sqrt{\Delta}}$. *If* $\kappa \geq n$, *then using the step size* $\gamma = 1/36L$, *PROXASAGA converges geometrically with rate factor* $\Omega(\frac{1}{\kappa})$. *If* $\kappa < n$, *then using the step size* $\gamma = 1/36n\mu$, *PROXASAGA converges geometrically with rate factor* $\Omega(\frac{1}{n})$. *In both cases, the convergence rate is the same as Sparse Proximal SAGA. Thus PROXASAGA is linearly faster than its sequential counterpart up to a constant factor. Note that in both cases the step size does not depend on* $\tau$.

*Furthermore, if* $\tau \leq 6\kappa$, *we can use a universal step size of* $\Theta(1/L)$ *to get a similar rate for PROXASAGA than Sparse Proximal SAGA, thus making it adaptive to local strong convexity since the knowledge of* $\kappa$ *is not required.*

These speedup regimes are comparable with the best ones obtained in the smooth case, including Niu et al. (2011); Reddi et al. (2015), even though unlike these papers, we support inconsistent reads and nonsmooth objective functions. The one exception is Leblond et al. (2017), where the authors prove that their algorithm, ASAGA, can obtain a linear speedup even without sparsity in the well-conditioned regime. In contrast, PROXASAGA always requires some sparsity. Whether this property for smooth objective functions could be extended to the composite case remains an open problem.

Relative to ASYSPCD, in the best case scenario (where the components of the gradient are uncorrelated, a somewhat unrealistic setting), ASYSPCD can get a near-linear speedup for $\tau$ as big as $\sqrt[4]{p}$. Our result states that $\tau = \mathcal{O}(1/\sqrt{\Delta})$ is necessary for a linear speedup. This means in case $\Delta \leq 1/\sqrt{p}$ our bound is better than the one obtained for ASYSPCD. Recalling that $1/n \leq \Delta \leq 1$, it appears that PROXASAGA is favored when $n$ is bigger than $\sqrt{p}$ whereas ASYSPCD may have a better bound otherwise, though this comparison should be taken with a grain of salt given the assumptions we had to make to arrive at comparable quantities. An extended comparison with the related work can be found in Appendix D.

## 4 Experiments

In this section, we compare PROXASAGA with related methods on different datasets. Although PROXASAGA can be applied more broadly, we focus on $\ell_1 + \ell_2$-regularized logistic regression, a model of particular practical importance. The objective function takes the form

$$\frac{1}{n} \sum_{i=1}^{n} \log\left(1 + \exp(-b_i \boldsymbol{a}_i^\intercal \boldsymbol{x})\right) + \frac{\lambda_1}{2}\|\boldsymbol{x}\|_2^2 + \lambda_2\|\boldsymbol{x}\|_1 \quad , \tag{6}$$

where $\boldsymbol{a}_i \in \mathbb{R}^p$ and $b_i \in \{-1, +1\}$ are the data samples. Following Defazio et al. (2014), we set $\lambda_1 = 1/n$. The amount of $\ell_1$ regularization ($\lambda_2$) is selected to give an approximate $1/10$ nonzero

Table 1: Description of datasets.

| Dataset | $n$ | $p$ | density | $L$ | $\Delta$ |
|---|---|---|---|---|---|
| **KDD 2010** (Yu et al., 2010) | 19,264,097 | 1,163,024 | $10^{-6}$ | 28.12 | 0.15 |
| **KDD 2012** (Juan et al., 2016) | 149,639,105 | 54,686,452 | $2 \times 10^{-7}$ | 1.25 | 0.85 |
| **Criteo** (Juan et al., 2016) | 45,840,617 | 1,000,000 | $4 \times 10^{-5}$ | 1.25 | 0.89 |

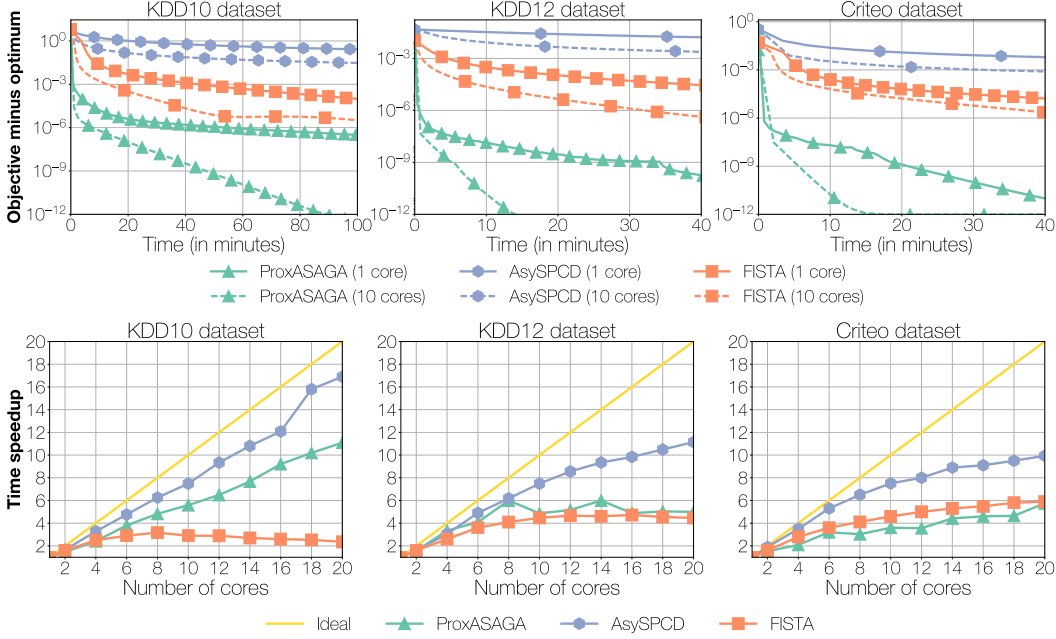

Figure 1: **Convergence for asynchronous stochastic methods for $\ell_1 + \ell_2$-regularized logistic regression**. Top: Suboptimality as a function of time for different asynchronous methods using 1 and 10 cores. Bottom: Running time speedup as function of the number of cores. PROXASAGA achieves significant speedups over its sequential version while being orders of magnitude faster than competing methods. ASYSPCD achieves the highest speedups but it also the slowest overall method.

coefficients. Implementation details are available in Appendix E. We chose the 3 datasets described in Table 1

**Results.** We compare three parallel asynchronous methods on the aforementioned datasets: PROX-ASAGA (this work),[6] ASYSPCD, the asynchronous proximal coordinate descent method of Liu & Wright (2015) and the (synchronous) FISTA algorithm (Beck & Teboulle, 2009), in which the gradient computation is parallelized by splitting the dataset into equal batches. We aim to benchmark these methods in the most realistic scenario possible; to this end we use the following step size: $1/2L$ for PROXASAGA, $1/L_c$ for ASYSPCD, where $L_c$ is the coordinate-wise Lipschitz constant of the gradient, while FISTA uses backtracking line-search. The results can be seen in Figure 1 (top) with both one (thus sequential) and ten processors. Two main observations can be made from this figure. First, PROXASAGA is significantly faster on these problems. Second, its asynchronous version offers a significant speedup over its sequential counterpart.

In Figure 1 (bottom) we present speedup with respect to the number of cores, where speedup is computed as the time to achieve a suboptimality of $10^{-10}$ with one core divided by the time to achieve the same suboptimality using several cores. While our *theoretical speedups* (with respect to the number of iterations) are almost linear as our theory predicts (see Appendix F), we observe a different story for our *running time* speedups. This can be attributed to memory access overhead, which our model does not take into account. As predicted by our theoretical results, we observe

a high correlation between the $\Delta$ dataset sparsity measure and the empirical speedup: KDD 2010 ($\Delta = 0.15$) achieves a 11x speedup, while in Criteo ($\Delta = 0.89$) the speedup is never above 6x.

Note that although competitor methods exhibit similar or sometimes better speedups, they remain orders of magnitude slower than PROXASAGA in running time for large sparse problems. In fact, our method is between 5x and 80x times faster (in time to reach $10^{-10}$ suboptimality) than FISTA and between 13x and 290x times faster than ASYSPCD (see Appendix F.3).

## 5  Conclusion and future work

In this work, we have described PROXASAGA, an asynchronous variance reduced algorithm with support for composite objective functions. This method builds upon a novel sparse variant of the (proximal) SAGA algorithm that takes advantage of sparsity in the individual gradients. We have proven that this algorithm is linearly convergent under a condition on the step size and that it is linearly faster than its sequential counterpart given a bound on the delay. Empirical benchmarks show that PROXASAGA is orders of magnitude faster than existing state-of-the-art methods.

This work can be extended in several ways. First, we have focused on the SAGA method as the basic iteration loop, but this approach can likely be extended to other proximal incremental schemes such as SGD or PROXSVRG. Second, as mentioned in §3.3, it is an open question whether it is possible to obtain convergence guarantees without any sparsity assumption, as was done for ASAGA.

## Acknowledgements

The authors would like to thank our colleagues Damien Garreau, Robert Gower, Thomas Kerdreux, Geoffrey Negiar, Konstantin Mishchenko and Kilian Fatras for their feedback on this manuscript, and Jean-Baptiste Alayrac for support managing the computational resources.

This work was partially supported by a Google Research Award. FP acknowledges support from the chaire *Économie des nouvelles données* with the *data science* joint research initiative with the *fonds AXA pour la recherche*.

## Footnotes

[2]Although some regularizers are sparsity inducing, large scale datasets are often extremely sparse and leveraging this property is crucial for the efficiency of the method.

[3]Since we have assumed that each individual $f_i$ is $L$-smooth, $f$ itself is $L$-smooth – but it could have a smaller smoothness constant. Our rates are in terms of this bigger $L/\mu$, as is standard in the SAGA literature.

[4]For example, in the `LibSVM` datasets suite, 8 out of the 11 datasets (as of May 2017) with more than a million samples have a density between $10^{-4}$ and $10^{-6}$.

[5]This is an extension of the framework of Niu et al. (2011), where consistent updates were assumed.

[6]A reference C++/Python implementation of is available at https://github.com/fabianp/ProxASAGA

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
