[Supplementary Material]

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

[7]The appearance of the sparsity constant $\Delta$ is coming from the crucial property that $\mathbf{E}\|\boldsymbol{x}\|_{(i)}^2 \leq \Delta\|\boldsymbol{x}\|^2$ $\forall x \in \mathbb{R}^p$ (see Eq. (39) in Leblond et al. (2017), where they use the notation $\| \cdot \|_i$ for our $\| \cdot \|_{(i)}$).

[8]Note that Leblond et al. (2017) analyzed the unconstrained scenario, and so $B_f(\hat{\boldsymbol{x}}_u,\boldsymbol{x}^*)$ is replaced by the simpler $f(\hat{x}_u) - f(\boldsymbol{x}^*)$ in their bound.

[9]For PROXASAGA the relevant quantity becomes the average number of features per data point. For ASYSPCD it is rather the average number of data points per feature. In both cases the tricks involved are not covered by the theory.

[10]To make sure $\tau$ is the same quantity for both algorithms, we have to assume that the iteration costs are homogeneous.

[11]To the best of our understanding, noting that extracting an interpretable bound from the given theoretical results was difficult. Furthermore, it appears that the proof technique may still have significant issues: for example, the "fully lock-free" assumption of Gu et al. (2016) allows for overwrites, and is thus incompatible with their framework of analysis, in particular their Eq. (8).

[12]https://www.csie.ntu.edu.tw/~cjlin/libsvmtools/datasets/

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

# Breaking the Nonsmooth Barrier: A Scalable Parallel Method for Composite Optimization

# Supplementary material

**Notations.** Throughout the supplementary material we use the following extra notation. We denote by $\langle \cdot, \cdot \rangle_{(i)}$ (resp. $\| \cdot \|_{(i)}$) the scalar product (resp. norm) restricted to blocks in $T_i$, i.e., $\langle \boldsymbol{x}, \boldsymbol{y} \rangle_{(i)} := \sum_{B \in T_i} \langle [\boldsymbol{x}]_B, [\boldsymbol{y}]_B \rangle$ and $\| \boldsymbol{x} \|_{(i)} := \sqrt{\langle \boldsymbol{x}, \boldsymbol{x} \rangle_{(i)}}$. We will also use the following definitions: $\varphi := \sum_{B \in \mathcal{B}} d_B h_B(\boldsymbol{x})$ and $\boldsymbol{D}$ is the diagonal matrix defined block-wise as $[\boldsymbol{D}]_{B,B} = d_B \boldsymbol{I}_{|B|}$.

The **Bregman divergence** associated with a convex function $f$ for points $\boldsymbol{x}, \boldsymbol{y}$ in its domain is defined as:

$$B_f(\boldsymbol{x}, \boldsymbol{y}) := f(\boldsymbol{x}) - f(\boldsymbol{y}) - \langle \nabla f(\boldsymbol{y}), \boldsymbol{x} - \boldsymbol{y} \rangle . \tag{7}$$

Note that the Bregman divergence is always positive due to the convexity of $f$.

## Appendix A   Basic properties

**Lemma 1.** *For any $\mu$-strongly convex function $f$ we have the following inequality:*

$$\langle \nabla f(\boldsymbol{y}) - \nabla f(\boldsymbol{x}), \boldsymbol{y} - \boldsymbol{x} \rangle \geq \frac{\mu}{2} \| \boldsymbol{y} - \boldsymbol{x} \|^2 + B_f(\boldsymbol{x}, \boldsymbol{y}) . \tag{8}$$

*Proof.* By strong convexity, $f$ verifies the inequality:

$$f(\boldsymbol{y}) \leq f(\boldsymbol{x}) + \langle \nabla f(\boldsymbol{y}), \boldsymbol{y} - \boldsymbol{x} \rangle - \frac{\mu}{2} \| \boldsymbol{y} - \boldsymbol{x} \|^2 , \tag{9}$$

for any $\boldsymbol{x}, \boldsymbol{y}$ in the domain (see e.g. (Nesterov, 2004)). We then have the equivalences:

$$
\begin{aligned}
& f(\boldsymbol{x}) \leq f(\boldsymbol{y}) + \langle \nabla f(\boldsymbol{x}), \boldsymbol{x} - \boldsymbol{y} \rangle - \frac{\mu}{2} \| \boldsymbol{x} - \boldsymbol{y} \|^2 \\
\iff & \frac{\mu}{2} \| \boldsymbol{x} - \boldsymbol{y} \|^2 + f(\boldsymbol{x}) - f(\boldsymbol{y}) \leq \langle \nabla f(\boldsymbol{x}), \boldsymbol{x} - \boldsymbol{y} \rangle \\
\iff & \frac{\mu}{2} \| \boldsymbol{x} - \boldsymbol{y} \|^2 + \underbrace{f(\boldsymbol{x}) - f(\boldsymbol{y}) - \langle \nabla f(\boldsymbol{y}), \boldsymbol{x} - \boldsymbol{y} \rangle}_{B_f(\boldsymbol{x}, \boldsymbol{y})} \leq \langle \nabla f(\boldsymbol{x}) - \nabla f(\boldsymbol{y}), \boldsymbol{x} - \boldsymbol{y} \rangle ,
\end{aligned}
\tag{10}
$$

where in the last line we have subtracted $\langle \nabla f(\boldsymbol{y}), \boldsymbol{x} - \boldsymbol{y} \rangle$ from both sides of the inequality. □

**Lemma 2.** *Let the $f_i$ be $L$-smooth and convex functions. Then it is verified that:*

$$\frac{1}{n} \sum_{i=1}^n \| \nabla f_i(\boldsymbol{x}) - \nabla f_i(\boldsymbol{y}) \|^2 \leq 2L B_f(\boldsymbol{x}, \boldsymbol{y}) . \tag{11}$$

*Proof.* Since each $f_i$ is $L$-smooth, it is verified (see e.g. Nesterov (2004, Theorem 2.1.5)) that

$$\| \nabla f_i(\boldsymbol{x}) - \nabla f_i(\boldsymbol{y}) \|^2 \leq 2L \big( f_i(\boldsymbol{x}) - f_i(\boldsymbol{y}) - \langle \nabla f_i(\boldsymbol{y}), \boldsymbol{x} - \boldsymbol{y} \rangle \big) . \tag{12}$$

The result is obtained by averaging over $i$. □

**Lemma 3** (Characterization of the proximal operator)**.** *Let $h$ be convex lower semicontinuous. Then we have the following characterization of the proximal operator:*

$$\boldsymbol{z} = \mathbf{prox}_{\gamma h}(\boldsymbol{x}) \iff \frac{1}{\gamma}(\boldsymbol{x} - \boldsymbol{z}) \in \partial h(\boldsymbol{z}) . \tag{13}$$

*Proof.* This is a direct consequence of the first order optimality conditions on the definition of proximal operator, see e.g. (Beck & Teboulle, 2009; Nesterov, 2013). □

**Lemma 4** (Firm non-expansiveness). *Let $\boldsymbol{x}, \tilde{\boldsymbol{x}}$ be two arbitrary vectors in $\mathbb{R}^p$ and $\boldsymbol{z}, \tilde{\boldsymbol{z}}$ be defined as $\boldsymbol{z} := \mathrm{prox}_{\gamma\varphi_i}(\boldsymbol{x})$, $\tilde{\boldsymbol{z}} := \mathrm{prox}_{\gamma\varphi_i}(\tilde{\boldsymbol{x}})$. Then it is verified that:*

$$\langle \boldsymbol{z} - \tilde{\boldsymbol{z}}, \boldsymbol{x} - \tilde{\boldsymbol{x}} \rangle_{(i)} \geq \|\boldsymbol{z} - \tilde{\boldsymbol{z}}\|_{(i)}^2 . \tag{14}$$

*Proof.* By the block-separability of $\varphi_i$, the proximal operator is the concatenation of the proximal operators of the blocks. In other words, for any block $B \in T_i$ we have:

$$[\boldsymbol{z}]_B = \mathbf{prox}_{\gamma\varphi_B}([\boldsymbol{x}]_B), \quad [\tilde{\boldsymbol{z}}]_B = \mathbf{prox}_{\gamma\varphi_B}([\tilde{\boldsymbol{x}}]_B), \tag{15}$$

where $\varphi_B$ is the restriction of $\varphi_i$ to $B$. By firm non-expansiveness of the proximal operator (see e.g. Bauschke & Combettes (2011, Proposition 4.2)) we have that:

$$\langle [\boldsymbol{z}]_B - [\tilde{\boldsymbol{z}}]_B, [\boldsymbol{x}]_B - [\tilde{\boldsymbol{x}}]_B \rangle \geq \|[\boldsymbol{z}]_B - [\tilde{\boldsymbol{z}}]_B\|^2 .$$

Summing over the blocks in $T_i$ yields the desired result. □

# Appendix B  Sparse Proximal SAGA

This Appendix contains all proofs for Section 2. The main result of this section is Theorem 1, whose proof is structured as follows:

- We start by proving four auxiliary results that will be used later on in the proofs of both synchronous and asynchronous variants. The first is the unbiasedness of key quantities used in the algorithm. The second is a characterization of the solutions of (OPT) in terms of $f$ and $\varphi$ (defined below) in Lemma 6. The third is a key inequality in Lemma 7 that relates the gradient mapping to other terms that arise in the optimization. The fourth is an upper bound on the variance terms of the gradient estimator, relating it to the Bregman divergence of $f$ and the past gradient estimator terms.

- In Lemma 9, we define an upper bound on the iterates $\|x_t - x^*\|^2$, called a Lyapunov function, and prove an inequality that relates this Lyapunov function value at the current iterate with its value at the previous iterate.

- Finally, in the proof of Theorem 1 we use the previous inequality in terms of the Lyapunov function to prove a geometric convergence of the iterates.

We start by proving the following unbiasedness result, mentioned in §2.

**Lemma 5.** *Let $\boldsymbol{D}_i$ and $\varphi_i$ be defined as in §2. Then it is verified that $\mathbf{E}\boldsymbol{D}_i = \boldsymbol{I}_p$ and $\mathbf{E}\,\varphi_i = h$.*

*Proof.* Let $B \in \mathcal{B}$ an arbitrary block. We have the following sequence of equalities:

$$\mathbf{E}[\boldsymbol{D}_i]_{B,B} = \frac{1}{n}\sum_{i=1}^{n}[\boldsymbol{D}_i]_{B,B} = \frac{1}{n}\sum_{i=1}^{n}d_B \mathbb{1}\{B \in T_i\}\boldsymbol{I}_{|B|} \tag{16}$$

$$= \frac{1}{n}\sum_{i=1}^{n}\frac{n}{n_B}\mathbb{1}\{B \in T_i\}\boldsymbol{I}_{|B|} \tag{17}$$

$$= \left(\frac{1}{n_B}\sum_{i=1}^{n}\mathbb{1}\{B \in T_i\}\right)\boldsymbol{I}_{|B|} = \boldsymbol{I}_{|B|}\,, \tag{18}$$

where the last equality comes from the definition of $n_B$. $\mathbf{E}\boldsymbol{D}_i = \boldsymbol{I}_p$ then follows from the arbitrariness of $B$.

Similarly, for $\varphi_i$ we have:

$$\mathbf{E}\varphi_i([\boldsymbol{x}]_B) = \frac{1}{n}\sum_{i=1}^{n}d_B \mathbb{1}\{B \in T_i\}h_B([\boldsymbol{x}]_B) \tag{19}$$

$$= \frac{1}{n}\sum_{i=1}^{n}\frac{n}{n_B}\mathbb{1}\{B \in T_i\}h_B([\boldsymbol{x}]_B) \tag{20}$$

$$= \left(\frac{1}{n_B}\sum_{i=1}^{n}\mathbb{1}\{B \in T_i\}\right)h_B([\boldsymbol{x}]_B) = h_B([\boldsymbol{x}]_B)\,, \tag{21}$$

Finally, the result $\mathbf{E}\,\varphi_i = h$ comes from adding over all blocks. $\qquad\square$

**Lemma 6.** *$x^*$ is a solution to (OPT) if and only if the following condition is verified:*

$$\boldsymbol{x}^* = \mathbf{prox}_{\gamma\varphi}\big(\boldsymbol{x}^* - \gamma\boldsymbol{D}\nabla f(\boldsymbol{x}^*)\big)\,. \tag{22}$$

*Proof.* By the first order optimality conditions, the solutions to (OPT) are characterized by the subdifferential inclusion $-\nabla f(\boldsymbol{x}^*) \in \partial h(\boldsymbol{x}^*)$. We can then write the following sequence of equiva-

lences:

$$-\nabla f(\boldsymbol{x}^*) \in \partial h(\boldsymbol{x}^*) \iff -\boldsymbol{D}\nabla f(\boldsymbol{x}^*) \in \boldsymbol{D}\partial h(\boldsymbol{x}^*) \tag{23}$$

(multiplying by $\boldsymbol{D}$, equivalence since diagonals are nonzero)

$$\iff -\boldsymbol{D}\nabla f(\boldsymbol{x}^*) \in \partial\varphi(\boldsymbol{x}^*) \tag{24}$$

(by definition of $\varphi$)

$$\iff \frac{1}{\gamma}(\boldsymbol{x}^* - \gamma\boldsymbol{D}\nabla f(\boldsymbol{x}^*) - \boldsymbol{x}^*) \in \partial\varphi(\boldsymbol{x}^*) \tag{25}$$

(adding and subtracting $\boldsymbol{x}^*$)

$$\iff \boldsymbol{x}^* = \mathbf{prox}_{\gamma\varphi}(\boldsymbol{x}^* - \gamma\boldsymbol{D}\nabla f(\boldsymbol{x}^*)). \tag{26}$$

(by Lemma 3)

Since all steps are equivalences, we have the desired result. $\qquad\square$

The following lemma will be key in the proof of convergence for both the sequential and the parallel versions of the algorithm. With this result, we will be able to bound the product between the gradient mapping and the iterate suboptimality by:

- First, the negative norm of the gradient mapping, which will be key in the parallel setting to cancel out the terms arising from the asynchrony.

- Second, variance terms in $\|\boldsymbol{v}_i - \boldsymbol{D}_i\nabla f(\boldsymbol{x}^*)\|^2$ that we will be able to bound by the Bregman divergence using Lemma 2.

- Third and last, a product with terms in $\langle \boldsymbol{v}_i - \boldsymbol{D}_i\nabla f(\boldsymbol{x}^*), \boldsymbol{x} - \boldsymbol{x}^* \rangle$, which taken in expectation gives $\langle \nabla f(\boldsymbol{x}) - \nabla f(\boldsymbol{x}^*), \boldsymbol{x} - \boldsymbol{x}^* \rangle$ and will allow us to apply Lemma 1 to obtain the contraction terms needed to obtain a geometric rate of convergence.

**Lemma 7** (Gradient mapping inequality). *Let $\boldsymbol{x}$ be an arbitrary vector, $\boldsymbol{x}^*$ a solution to (OPT), $\boldsymbol{v}_i$ as defined in (SPS) and $\boldsymbol{g} = g(\boldsymbol{x}, \boldsymbol{v}_i, i)$ the gradient mapping defined in (3). Then the following inequality is verified for any $\beta > 0$:*

$$\langle \boldsymbol{g}, \boldsymbol{x} - \boldsymbol{x}^* \rangle \geq -\frac{\gamma}{2}(\beta - 2)\|\boldsymbol{g}\|^2 - \frac{\gamma}{2\beta}\|\boldsymbol{v}_i - \boldsymbol{D}_i\nabla f(\boldsymbol{x}^*)\|^2 + \langle \boldsymbol{v}_i - \boldsymbol{D}_i\nabla f(\boldsymbol{x}^*), \boldsymbol{x} - \boldsymbol{x}^* \rangle. \tag{27}$$

*Proof.* By firm non-expansiveness of the proximal operator (Lemma 4) applied to $\boldsymbol{z} = \mathbf{prox}_{\gamma\varphi_i}(\boldsymbol{x} - \gamma\boldsymbol{v}_i)$ and $\tilde{\boldsymbol{z}} = \mathbf{prox}_{\gamma\varphi_i}(\boldsymbol{x}^* - \gamma\boldsymbol{D}\nabla f(\boldsymbol{x}^*))$ we have:

$$\|\boldsymbol{z} - \tilde{\boldsymbol{z}}\|_{(i)}^2 - \langle \boldsymbol{z} - \tilde{\boldsymbol{z}}, \boldsymbol{x} - \gamma\boldsymbol{v}_i - \boldsymbol{x}^* + \gamma\boldsymbol{D}\nabla f(\boldsymbol{x}^*) \rangle_{(i)} \leq 0. \tag{28}$$

By the (SPS) iteration we have $\boldsymbol{x}^+ = \boldsymbol{z}$ and by Lemma 3 we have that $[\boldsymbol{z}]_{T_i} = [\boldsymbol{x}^*]_{T_i}$, hence the above can be rewritten as

$$\|\boldsymbol{x}^+ - \boldsymbol{x}^*\|_{(i)}^2 - \langle \boldsymbol{x}^+ - \boldsymbol{x}^*, \boldsymbol{x} - \gamma\boldsymbol{v}_i - \boldsymbol{x}^* + \gamma\boldsymbol{D}\nabla f(\boldsymbol{x}^*) \rangle_{(i)} \leq 0. \tag{29}$$

We can now write the following sequence of inequalities

$$\langle \gamma \boldsymbol{g}, \boldsymbol{x} - \boldsymbol{x}^* \rangle = \langle \boldsymbol{x} - \boldsymbol{x}^+, \boldsymbol{x} - \boldsymbol{x}^* \rangle_{(i)} \qquad \text{(by definition and sparsity of } g) \tag{30}$$

$$= \langle \boldsymbol{x} - \boldsymbol{x}^+ + \boldsymbol{x}^* - \boldsymbol{x}^*, \boldsymbol{x} - \boldsymbol{x}^* \rangle_{(i)} \tag{31}$$

$$= \|\boldsymbol{x} - \boldsymbol{x}^*\|_{(i)}^2 - \langle \boldsymbol{x}^+ - \boldsymbol{x}^*, \boldsymbol{x} - \boldsymbol{x}^* \rangle_{(i)} \tag{32}$$

$$\geq \|\boldsymbol{x} - \boldsymbol{x}^*\|_{(i)}^2 - \langle \boldsymbol{x}^+ - \boldsymbol{x}^*, 2\boldsymbol{x} - \gamma \boldsymbol{v}_i - 2\boldsymbol{x}^* + \gamma \boldsymbol{D}\nabla f(\boldsymbol{x}^*) \rangle_{(i)} + \|\boldsymbol{x}^+ - \boldsymbol{x}^*\|_{(i)}^2 \tag{33}$$
$$\text{(adding Eq. (29))}$$

$$= \|\boldsymbol{x} - \boldsymbol{x}^+\|_{(i)}^2 + \langle \boldsymbol{x}^+ - \boldsymbol{x}^*, \gamma \boldsymbol{v}_i - \gamma \boldsymbol{D}\nabla f(\boldsymbol{x}^*) \rangle_{(i)} \qquad \text{(completing the square)} \tag{34}$$

$$= \|\boldsymbol{x} - \boldsymbol{x}^+\|_{(i)}^2 + \langle \boldsymbol{x} - \boldsymbol{x}^*, \gamma \boldsymbol{v}_i - \gamma \boldsymbol{D}\nabla f(\boldsymbol{x}^*) \rangle_{(i)} - \langle \boldsymbol{x} - \boldsymbol{x}^+, \gamma \boldsymbol{v}_i - \gamma \boldsymbol{D}\nabla f(\boldsymbol{x}^*) \rangle_{(i)} \tag{35}$$
$$\text{(adding and substracting } \boldsymbol{x})$$

$$\geq \left(1 - \frac{\beta}{2}\right)\|\boldsymbol{x} - \boldsymbol{x}^+\|_{(i)}^2 - \frac{\gamma^2}{2\beta}\|\boldsymbol{v}_i - \boldsymbol{D}\nabla f(\boldsymbol{x}^*)\|_{(i)}^2 + \gamma\langle \boldsymbol{v}_i - \boldsymbol{D}\nabla f(\boldsymbol{x}^*), \boldsymbol{x} - \boldsymbol{x}^* \rangle_{(i)} \tag{36}$$

$$\text{(Young's inequality } 2\langle a, b \rangle \leq \frac{\|a\|^2}{\beta} + \beta\|b\|^2 \text{, valid for arbitrary } \beta > 0)$$

$$\geq \left(1 - \frac{\beta}{2}\right)\|\boldsymbol{x} - \boldsymbol{x}^+\|_{(i)}^2 - \frac{\gamma^2}{2\beta}\|\boldsymbol{v}_i - \boldsymbol{D}_i\nabla f(\boldsymbol{x}^*)\|^2 + \gamma\langle \boldsymbol{v}_i - \boldsymbol{D}_i\nabla f(\boldsymbol{x}^*), \boldsymbol{x} - \boldsymbol{x}^* \rangle \tag{37}$$
$$\text{(by definition of } \boldsymbol{D}_i \text{ and using the fact that } \boldsymbol{v}_i \text{ is } T_i\text{-sparse)}$$

$$= \left(1 - \frac{\beta}{2}\right)\|\gamma \boldsymbol{g}\|^2 - \frac{\gamma^2}{2\beta}\|\boldsymbol{v}_i - \boldsymbol{D}_i\nabla f(\boldsymbol{x}^*)\|^2 + \gamma\langle \boldsymbol{v}_i - \boldsymbol{D}\nabla f(\boldsymbol{x}^*), \boldsymbol{x} - \boldsymbol{x}^* \rangle, \tag{38}$$

where in the last inequality we have used the fact that $\boldsymbol{g}$ is $T_i$-sparse. Finally, dividing by $\gamma$ both sides yields the desired result. $\qquad \square$

**Lemma 8** (Upper bound on the gradient estimator variance). *For arbitrary vectors $\boldsymbol{x}$, $(\boldsymbol{\alpha}_i)_{i=0}^n$, and $\boldsymbol{v}_i$ as defined in* (SPS) *we have:*

$$\mathbf{E}\|\boldsymbol{v}_i - \boldsymbol{D}_i\nabla f(\boldsymbol{x}^*)\|^2 \leq 4LB_f(\boldsymbol{x}, \boldsymbol{x}^*) + 2\mathbf{E}\|\boldsymbol{\alpha}_i - \nabla f_i(\boldsymbol{x}^*)\|^2. \tag{39}$$

*Proof.* We will now bound the variance terms. For this we have:

$$\mathbf{E}\|\boldsymbol{v}_i - \boldsymbol{D}\nabla f(\boldsymbol{x}^*)\|_{(i)}^2 = \mathbf{E}\|\nabla f_i(\boldsymbol{x}) - \nabla f_i(\boldsymbol{x}^*) + \nabla f_i(\boldsymbol{x}^*) - \boldsymbol{\alpha}_i + \boldsymbol{D}_i\overline{\boldsymbol{\alpha}} - \boldsymbol{D}\nabla f(\boldsymbol{x}^*)\|_{(i)}^2 \tag{40}$$

$$\leq 2\mathbf{E}\|\nabla f_i(\boldsymbol{x}) - \nabla f_i(\boldsymbol{x}^*)\|^2 + 2\mathbf{E}\|\nabla f_i(\boldsymbol{x}^*) - \boldsymbol{\alpha}_i - (\boldsymbol{D}\nabla f(\boldsymbol{x}^*) - \boldsymbol{D}\overline{\boldsymbol{\alpha}})\|_{(i)}^2 \tag{41}$$
$$\text{(by inequality } \|a + b\|^2 \leq 2\|a\|^2 + 2\|b\|^2)$$

$$= 2\mathbf{E}\|\nabla f_i(\boldsymbol{x}) - \nabla f_i(\boldsymbol{x}^*)\|^2 + 2\mathbf{E}\|\nabla f_i(\boldsymbol{x}^*) - \boldsymbol{\alpha}_i\|^2$$
$$- 4\mathbf{E}\langle \nabla f_i(\boldsymbol{x}^*) - \boldsymbol{\alpha}_i, \boldsymbol{D}\nabla f(\boldsymbol{x}^*) - \boldsymbol{D}\overline{\boldsymbol{\alpha}} \rangle_{(i)} + 2\mathbf{E}\|\boldsymbol{D}\nabla f(\boldsymbol{x}^*) - \boldsymbol{D}\overline{\boldsymbol{\alpha}}\|_{(i)}^2. \tag{42}$$
$$\text{(developing the square)}$$

We will now simplify the last two terms in the above expression. For the first of the two last terms we have:

$$-4\mathbf{E}\langle \nabla f_i(\boldsymbol{x}^*) - \boldsymbol{\alpha}_i, \boldsymbol{D}\nabla f(\boldsymbol{x}^*) - \boldsymbol{D}\overline{\boldsymbol{\alpha}} \rangle_{(i)} = -4\mathbf{E}\langle \nabla f_i(\boldsymbol{x}^*) - \boldsymbol{\alpha}_i, \boldsymbol{D}\nabla f(\boldsymbol{x}^*) - \boldsymbol{D}\overline{\boldsymbol{\alpha}} \rangle \tag{43}$$
$$\text{(since the support of first term is contained in } T_i)$$

$$= -4\langle \nabla f(\boldsymbol{x}^*) - \overline{\boldsymbol{\alpha}}, \boldsymbol{D}\nabla f(\boldsymbol{x}^*) - \boldsymbol{D}\overline{\boldsymbol{\alpha}} \rangle \tag{44}$$
$$\text{(taking expectations)} \tag{45}$$

$$= -4\|\nabla f(\boldsymbol{x}^*) - \overline{\boldsymbol{\alpha}}\|_{\boldsymbol{D}}^2. \tag{46}$$

Similarly, for the last term we have:

$$2\mathbf{E}\|\boldsymbol{D}\nabla f(\boldsymbol{x}^*) - \boldsymbol{D}\overline{\boldsymbol{\alpha}}\|_{(i)}^2 = 2\mathbf{E}\langle \boldsymbol{D}_i\nabla f(\boldsymbol{x}^*) - \boldsymbol{D}_i\overline{\boldsymbol{\alpha}}, \boldsymbol{D}\nabla f(\boldsymbol{x}^*) - \boldsymbol{D}\overline{\boldsymbol{\alpha}} \rangle \tag{47}$$

$$= 2\langle \nabla f(\boldsymbol{x}^*) - \overline{\boldsymbol{\alpha}}, \boldsymbol{D}\nabla f(\boldsymbol{x}^*) - \boldsymbol{D}\overline{\boldsymbol{\alpha}} \rangle \tag{48}$$
$$\text{(using Lemma 5)}$$

$$= 2\|\nabla f(\boldsymbol{x}^*) - \overline{\boldsymbol{\alpha}}\|_{\boldsymbol{D}}^2. \tag{49}$$

and so the addition of these terms is negative and can be dropped. In all, for the variance terms we have

$$\mathbf{E}\|\boldsymbol{v}_i - \boldsymbol{D}\nabla f(\boldsymbol{x}^*)\|_{(i)}^2 \leq 2\mathbf{E}\|\nabla f_i(\boldsymbol{x}) - \nabla f_i(\boldsymbol{x}^*)\|^2 + 2\mathbf{E}\|\boldsymbol{\alpha}_i - \nabla f_i(\boldsymbol{x}^*)\|^2 \tag{50}$$

$$\leq 4LB_f(\boldsymbol{x}, \boldsymbol{x}^*) + 2\mathbf{E}\|\boldsymbol{\alpha}_i - \nabla f_i(\boldsymbol{x}^*)\|^2. \qquad \text{(by Lemma 2)} \tag{51}$$

□

We now define an upper bound on the quantity that we would like to bound, often called a Lyapunov function, and establish a recursive inequality on this Lyapunov function.

**Lemma 9** (Lyapunov inequality). *Let $\mathcal{L}$ be the following $c$-parametrized function:*

$$\mathcal{L}(\boldsymbol{x}, \boldsymbol{\alpha}) := \|\boldsymbol{x} - \boldsymbol{x}^*\|^2 + \frac{c}{n}\sum_{i=1}^{n}\|\boldsymbol{\alpha}_i - \nabla f_i(\boldsymbol{x}^*)\|^2. \tag{52}$$

*Let $\boldsymbol{x}^+$ and $\boldsymbol{\alpha}^+$ be obtained from the Sparse Proximal SAGA updates (SPS). Then we have:*

$$\mathbf{E}\mathcal{L}(\boldsymbol{x}^+, \boldsymbol{\alpha}^+) - \mathcal{L}(\boldsymbol{x}, \boldsymbol{\alpha}) \leq -\gamma\mu\|\boldsymbol{x} - \boldsymbol{x}^*\|^2 + \left(4L\gamma^2 - 2\gamma + 2L\frac{c}{n}\right)B_f(\boldsymbol{x}, \boldsymbol{x}^*)$$
$$+ \left(2\gamma^2 - \frac{c}{n}\right)\mathbf{E}\|\boldsymbol{\alpha}_i - \nabla f_i(\boldsymbol{x})\|^2. \tag{53}$$

*Proof.* For the first term of $\mathcal{L}$ we have:

$$\|\boldsymbol{x}^+ - \boldsymbol{x}^*\|^2 = \|\boldsymbol{x} - \gamma\boldsymbol{g} - \boldsymbol{x}^*\|^2 \qquad (\boldsymbol{g} := \boldsymbol{g}(\boldsymbol{x}, \boldsymbol{v}_i, i)) \tag{54}$$

$$= \|\boldsymbol{x} - \boldsymbol{x}^*\|^2 - 2\gamma\langle\boldsymbol{g}, \boldsymbol{x} - \boldsymbol{x}^*\rangle + \|\gamma\boldsymbol{g}\|^2 \tag{55}$$

$$\leq \|\boldsymbol{x} - \boldsymbol{x}^*\|^2 + \gamma^2\|\boldsymbol{v}_i - \boldsymbol{D}_i\nabla f(\boldsymbol{x}^*)\|^2 - 2\gamma\langle\boldsymbol{v}_i - \boldsymbol{D}_i\nabla f(\boldsymbol{x}^*), \boldsymbol{x} - \boldsymbol{x}^*\rangle \tag{56}$$
$$\text{(by Lemma 7 with } \beta = 1)$$

Since $\boldsymbol{v}_i$ is an unbiased estimator of the gradient and $\mathbf{E}\boldsymbol{D}_i = \boldsymbol{I}_p$, taking expectations we have:

$$\mathbf{E}\|\boldsymbol{x}^+ - \boldsymbol{x}^*\|^2 \leq \|\boldsymbol{x} - \boldsymbol{x}^*\|^2 + \gamma^2\mathbf{E}\|\boldsymbol{v}_i - \boldsymbol{D}_i\nabla f(\boldsymbol{x}^*)\|^2 - 2\gamma\langle\nabla f(\boldsymbol{x}) - \nabla f(\boldsymbol{x}^*), \boldsymbol{x} - \boldsymbol{x}^*\rangle$$
$$\leq (1 - \gamma\mu)\|\boldsymbol{x} - \boldsymbol{x}^*\|^2 + \gamma^2\mathbf{E}\|\boldsymbol{v}_i - \boldsymbol{D}_i\nabla f(\boldsymbol{x}^*)\|^2 - 2\gamma B_f(\boldsymbol{x}, \boldsymbol{x}^*). \tag{57}$$
$$\text{(by Lemma 1)}$$

By using the variance terms bound (Lemma 8) in the previous equation we have:

$$\mathbf{E}\|\boldsymbol{x}^+ - \boldsymbol{x}^*\|^2 \leq (1 - \gamma\mu)\|\boldsymbol{x} - \boldsymbol{x}^*\|^2 + (4L\gamma^2 - 2\gamma)B_f(\boldsymbol{x}, \boldsymbol{x}^*)$$
$$+ 2\gamma^2\mathbf{E}\|\boldsymbol{\alpha}_i - \nabla f_i(\boldsymbol{x}^*)\|^2. \tag{58}$$

We will now bound the second term of the Lyapunov function. For a fixed index $j$ we have $\mathbf{E}\|\boldsymbol{\alpha}_j^+ - \nabla f_j(\boldsymbol{x}^*)\|^2 = (1 - \frac{1}{n})\|\boldsymbol{\alpha}_j - \nabla f_j(\boldsymbol{x}^*)\|^2 + \frac{1}{n}\|\nabla f_j(\boldsymbol{x}) - \nabla f_j(\boldsymbol{x}^*)\|^2$ since each $\boldsymbol{\alpha}_j$ has probability $\frac{1}{n}$ of being updated at a given iteration. Summing over all indices we obtain:

$$\mathbf{E}\left[\frac{1}{n}\sum_{i=1}^{n}\|\boldsymbol{\alpha}_i^+ - \nabla f_i(\boldsymbol{x}^*)\|^2\right] = \left(1 - \frac{1}{n}\right)\frac{1}{n}\sum_{i=1}^{n}\|\boldsymbol{\alpha}_i - \nabla f_i(\boldsymbol{x}^*)\|^2 \tag{59}$$

$$+ \frac{1}{n^2}\sum_{i=1}^{n}\|\nabla f_i(\boldsymbol{x}) - \nabla f_i(\boldsymbol{x}^*)\|^2 \tag{60}$$

$$\leq \left(1 - \frac{1}{n}\right)\frac{1}{n}\sum_{i=1}^{n}\|\boldsymbol{\alpha}_i - \nabla f_i(\boldsymbol{x}^*)\|^2 + \frac{2}{n}LB_f(\boldsymbol{x}, \boldsymbol{x}^*) \tag{61}$$

$$\text{(by Lemma 2)}$$

$$= \left(1 - \frac{1}{n}\right)\mathbf{E}\|\boldsymbol{\alpha}_i - \nabla f_i(\boldsymbol{x}^*)\|^2 + \frac{2}{n}LB_f(\boldsymbol{x}, \boldsymbol{x}^*). \tag{62}$$

Combining Eq. (58) and (61) we have:

$$\mathbf{E}\mathcal{L}(\boldsymbol{x}^+, \boldsymbol{\alpha}^+) \leq (1 - \gamma\mu)\|\boldsymbol{x} - \boldsymbol{x}^*\|^2 + (4L\gamma^2 - 2\gamma)B_f(\boldsymbol{x}, \boldsymbol{x}^*) + 2\gamma^2\mathbf{E}\|\boldsymbol{\alpha}_i - \nabla f_i(\boldsymbol{x}^*)\|^2$$

$$+ c\left[\left(1 - \frac{1}{n}\right)\mathbf{E}\|\boldsymbol{\alpha}_i - \nabla f_i(\boldsymbol{x}^*)\|^2 + \frac{1}{n}2LB_f(\boldsymbol{x}, \boldsymbol{x}^*)\right] \tag{63}$$

$$= (1 - \gamma\mu)\|\boldsymbol{x} - \boldsymbol{x}^*\|^2 + \left(4L\gamma^2 - 2\gamma + 2L\frac{c}{n}\right)B_f(\boldsymbol{x}, \boldsymbol{x}^*)$$

$$+ \left(2\gamma^2 - \frac{c}{n}\right)\mathbf{E}\|\boldsymbol{\alpha}_i - \nabla f_i(\boldsymbol{x}^*)\|^2 + c\mathbf{E}\|\boldsymbol{\alpha}_i - \nabla f_i(\boldsymbol{x}^*)\|^2 \tag{64}$$

$$= \mathcal{L}(\boldsymbol{x}, \boldsymbol{\alpha}) - \gamma\mu\|\boldsymbol{x} - \boldsymbol{x}^*\|^2 + \left(4L\gamma^2 - 2\gamma + 2L\frac{c}{n}\right)B_f(\boldsymbol{x}, \boldsymbol{x}^*)$$

$$+ \left(2\gamma^2 - \frac{c}{n}\right)\mathbf{E}\|\boldsymbol{\alpha}_i - \nabla f_i(\boldsymbol{x}^*)\|^2. \tag{65}$$

Finally, subtracting $\mathcal{L}(\boldsymbol{x}, \boldsymbol{\alpha})$ from both sides yields the desired result. $\qquad\square$

**Theorem 1.** *Let $\gamma = \frac{a}{5L}$ for any $a \leq 1$ and $f$ be $\mu$-strongly convex. Then Sparse Proximal* SAGA *converges geometrically in expectation with a rate factor of at least $\rho = \frac{1}{5}\min\{\frac{1}{n}, a\frac{1}{\kappa}\}$. That is, for $\boldsymbol{x}_t$ obtained after $t$ updates and $\boldsymbol{x}^*$ the solution to* (OPT), *we have the bound:*

$$\mathbb{E}\|\boldsymbol{x}_t - \boldsymbol{x}^*\|^2 \leq (1 - \rho)^t C_0, \quad \text{with } C_0 := \|\boldsymbol{x}_0 - \boldsymbol{x}^*\|^2 + \frac{1}{5L^2}\sum_{i=1}^{n}\|\boldsymbol{\alpha}_i^0 - \nabla f_i(\boldsymbol{x}^*)\|^2 \quad .$$

*Proof.* Let $\overline{H} := \frac{1}{n}\sum_i\|\boldsymbol{\alpha}_i - \nabla f_i(\boldsymbol{x}^*)\|^2$. By the Lyapunov inequality from Lemma 9, we have:

$$\mathbf{E}\mathcal{L}_{t+1} - (1 - \rho)\mathcal{L}_t \leq \rho\mathcal{L}_t - \gamma\mu\|\boldsymbol{x}_t - \boldsymbol{x}^*\|^2 + \left(4L\gamma^2 - 2\gamma + 2L\frac{c}{n}\right)B_f(\boldsymbol{x}_t, \boldsymbol{x}^*) + \left(2\gamma^2 - \frac{c}{n}\right)\overline{H}$$

$$= (\rho - \gamma\mu)\|\boldsymbol{x}_t - \boldsymbol{x}^*\|^2 + \left(4L\gamma^2 - 2\gamma + 2L\frac{c}{n}\right)B_f(\boldsymbol{x}_t, \boldsymbol{x}^*) + \left[2\gamma^2 + c\left(\rho - \frac{1}{n}\right)\right]\overline{H}$$

(by definition of $\mathcal{L}_t$)

$$\leq (\rho - \gamma\mu)\|\boldsymbol{x}_t - \boldsymbol{x}^*\|^2 + \left(4L\gamma^2 - 2\gamma + 2L\frac{c}{n}\right)B_f(\boldsymbol{x}_t, \boldsymbol{x}^*) + \left(2\gamma^2 - \frac{2c}{3n}\right)\overline{H} \tag{66}$$

(choosing $\rho \leq \frac{1}{3n}$)

$$= (\rho - \gamma\mu)\|\boldsymbol{x}_t - \boldsymbol{x}^*\|^2 + \left(10L\gamma^2 - 2\gamma\right)B_f(\boldsymbol{x}_t, \boldsymbol{x}^*) \tag{67}$$

(choosing $\frac{c}{n} = 3\gamma^2$)

$$\leq \left(\rho - \frac{a\mu}{5L}\right)\|\boldsymbol{x}_t - \boldsymbol{x}^*\|^2 \qquad \text{(for all } \gamma = \frac{a}{5L}, a \leq 1) \tag{68}$$

$$\leq 0. \qquad \text{(for } \rho \leq \frac{a}{5} \cdot \frac{\mu}{L}) \tag{69}$$

And so we have the bound:

$$\mathbf{E}\mathcal{L}_{t+1} \leq \left(1 - \min\left\{\frac{1}{3n}, \frac{a}{5} \cdot \frac{1}{\kappa}\right\}\right)\mathcal{L}_t \leq \left(1 - \frac{1}{5}\min\left\{\frac{1}{n}, a \cdot \frac{1}{\kappa}\right\}\right)\mathcal{L}_t, \tag{70}$$

where in the last inequality we have used the trivial bound $\frac{1}{3n} \leq \frac{1}{5n}$ merely for clarity of exposition. Chaining expectations from $t$ to $0$ we have:

$$\mathbb{E}\mathcal{L}_{t+1} \leq \left(1 - \frac{1}{5}\min\left\{\frac{1}{n}, a \cdot \frac{1}{\kappa}\right\}\right)^{t+1}\mathcal{L}_0 \tag{71}$$

$$= \left(1 - \frac{1}{5}\min\left\{\frac{1}{n}, a \cdot \frac{1}{\kappa}\right\}\right)^{t+1}\left(\|\boldsymbol{x}_0 - \boldsymbol{x}^*\|^2 + \frac{3a^2}{5^2L^2}\sum_{i=1}^{n}\|\boldsymbol{\alpha}_i^0 - \nabla f_i(\boldsymbol{x}^*)\|^2\right) \tag{72}$$

$$\leq \left(1 - \frac{1}{5}\min\left\{\frac{1}{n}, a \cdot \frac{1}{\kappa}\right\}\right)^{t+1}\left(\|\boldsymbol{x}_0 - \boldsymbol{x}^*\|^2 + \frac{1}{5L^2}\sum_{i=1}^{n}\|\boldsymbol{\alpha}_i^0 - \nabla f_i(\boldsymbol{x}^*)\|^2\right) \tag{73}$$

(since $a \leq 1$ and $3/5 \leq 1$).

The fact that $\mathcal{L}_t$ is a majorizer of $\|\boldsymbol{x}_t - \boldsymbol{x}^*\|^2$ completes the proof. $\qquad\square$

# Appendix C    ProxASAGA

In this Appendix we provide the proofs for results from Section 3, that is Theorem 2 (the convergence theorem for PROXASAGA) and Corollary 1 (its speedup result).

**Notation.**    Through this section, we use the following shorthand for the gradient mapping: $\boldsymbol{g}_t := \boldsymbol{g}(\hat{\boldsymbol{x}}_t, \hat{\boldsymbol{v}}^t_{i_t}, i_t)$.

## Appendix C.1    Proof outline.

As in the smooth case ($h = 0$), we start by using the definition of $\boldsymbol{x}_{t+1}$ in Eq. (4) to relate the distance to the optimum in terms of its previous iterates:

$$\|\boldsymbol{x}_{t+1} - \boldsymbol{x}^*\|^2 = \|\boldsymbol{x}_t - \boldsymbol{x}^*\|^2 + 2\gamma\langle\hat{\boldsymbol{x}}_t - \boldsymbol{x}_t, \boldsymbol{g}_t\rangle + \gamma^2\|\boldsymbol{g}_t\|^2 - 2\gamma\langle\hat{\boldsymbol{x}}_t - \boldsymbol{x}^*, \boldsymbol{g}_t\rangle. \tag{74}$$

However, in this case $\boldsymbol{g}_t$ is not a gradient estimator but a gradient mapping, so we cannot continue as is customary – by using the unbiasedness of the gradient in the $\langle\hat{\boldsymbol{x}}_t - \boldsymbol{x}^*, \boldsymbol{g}_t\rangle$ term together with the strong convexity of $f$ (see Leblond et al. (2017, Section 3.5)).

To circumvent this difficulty, we derive a tailored inequality for the gradient mapping (Lemma 7 in Appendix B), which in turn allows us to use the classical unbiasedness and strong convexity arguments to get the following inequality:

$$a_{t+1} \le (1 - \frac{\gamma\mu}{2})a_t + \gamma^2\mathbb{E}\|\boldsymbol{g}_t\|^2 - 2\gamma\mathbb{E}B_f(\hat{\boldsymbol{x}}_t, \boldsymbol{x}^*) + \underbrace{\gamma\mu\mathbb{E}\|\hat{\boldsymbol{x}}_t - \boldsymbol{x}\|^2 + 2\gamma\mathbb{E}\langle\boldsymbol{g}_t, \hat{\boldsymbol{x}}_t - \boldsymbol{x}_t\rangle}_{\text{additional asynchrony terms}} \tag{75}$$

$$\underbrace{+\gamma^2(\beta - 2)\mathbb{E}\|\boldsymbol{g}_t\|^2 + \frac{\gamma^2}{\beta}\mathbb{E}\|\hat{\boldsymbol{v}}^t_{i_t} - \boldsymbol{D}_{i_t}\nabla f(\boldsymbol{x}^*)\|^2}_{\text{additional proximal and variance terms}},$$

where $a_t := \mathbb{E}\|\boldsymbol{x}_t - \boldsymbol{x}^*\|^2$. Note that since $f$ is strongly convex, $B_f(\hat{\boldsymbol{x}}_t, \boldsymbol{x}^*) \ge \frac{\mu}{2}\|\hat{\boldsymbol{x}}_t - \boldsymbol{x}^*\|^2$.

In the smooth setting, one first expresses the additional asynchrony terms as linear combinations of past gradient variance terms $(\mathbb{E}\|\boldsymbol{g}_u\|^2)_{0 \le u \le t}$. Then one crucially uses the negative Bregman divergence term to control the variance terms. However, in our current setting, we cannot relate the norm of the gradient mapping $\mathbb{E}\|\boldsymbol{g}_t\|^2$ to the Bregman divergence (from which $h$ is absent). Instead, we use the negative term $\gamma^2(\beta - 1)\mathbb{E}\|\boldsymbol{g}_t\|^2$ to control all the $(\mathbb{E}\|\boldsymbol{g}_u\|^2)_{0 \le u \le t}$ terms that arise from asynchrony.

The rest of the proof consists in:

*i*) expressing the additional asynchrony terms as linear combinations of $(\mathbb{E}\|\boldsymbol{g}_u\|^2)_{0 \le u \le t}$, following Leblond et al. (2017, Lemma 1);

*ii*) expressing the last variance term, $\|\hat{\boldsymbol{v}}^t_{i_t} - D_{i_t}\nabla f(\boldsymbol{x}^*)\|^2$, as a linear combination of past Bregman divergences (Lemma 8 in Appendix B and Lemma 2 from Leblond et al. (2017));

*iii*) defining a Lyapunov function, $\mathcal{L}_t := \sum_{u=0}^t (1 - \rho)^{t-u} a_u$, and proving that it is bounded by a contraction given conditions on the maximum step size and delay.

## Appendix C.2    Detailed proof

**Theorem 2** (Convergence guarantee and rate of PROXASAGA). *Suppose $\tau \le \frac{1}{10\sqrt{\Delta}}$. For any step size $\gamma = \frac{a}{L}$ with $a \le \frac{1}{36}\min\{1, \frac{6\kappa}{\tau}\}$, the inconsistent read iterates of Algorithm 1 converge in expectation at a geometric rate factor of at least: $\rho(a) = \frac{1}{5}\min\left\{\frac{1}{n}, a\frac{1}{\kappa}\right\}$, i.e. $\mathbb{E}\|\hat{\boldsymbol{x}}_t - \boldsymbol{x}^*\|^2 \le (1 - \rho)^t \tilde{C}_0$, where $\tilde{C}_0$ is a constant independent of $t$ ($\approx \frac{n\kappa}{a}C_0$ with $C_0$ as defined in Theorem ??).*

*Proof.* In order to get an **initial recursive inequality**, we first unroll the (virtual) update:

$$\|\boldsymbol{x}_{t+1} - \boldsymbol{x}^*\|^2 = \|\boldsymbol{x}_t - \gamma\boldsymbol{g}_t - \boldsymbol{x}^*\|^2 = \|\boldsymbol{x}_t - \boldsymbol{x}^*\|^2 + \|\gamma\boldsymbol{g}_t\|^2 - 2\gamma\langle\boldsymbol{g}_t, \boldsymbol{x}_t - \boldsymbol{x}^*\rangle$$

$$= \|\boldsymbol{x}_t - \boldsymbol{x}^*\|^2 + \|\gamma\boldsymbol{g}_t\|^2 - 2\gamma\langle\boldsymbol{g}_t, \hat{\boldsymbol{x}}_t - \boldsymbol{x}^*\rangle + 2\gamma\langle\boldsymbol{g}_t, \hat{\boldsymbol{x}}_t - \boldsymbol{x}_t\rangle, \tag{76}$$

and then apply Lemma 7 with $\boldsymbol{x} = \hat{\boldsymbol{x}}_t$ and $\boldsymbol{v} = \hat{\boldsymbol{v}}_{i_t}^t$. Note that in this case we have $\boldsymbol{g} = \boldsymbol{g}_t$ and $\langle \cdot \rangle_{(i)} = \langle \cdot \rangle_{(i_t)}$.

$$
\begin{aligned}
\|\boldsymbol{x}_{t+1} - \boldsymbol{x}^*\|^2 &\leq \|\boldsymbol{x}_t - \boldsymbol{x}^*\|^2 + 2\gamma\langle \boldsymbol{g}_t, \hat{\boldsymbol{x}}_t - \boldsymbol{x}_t \rangle + \gamma^2\|\boldsymbol{g}_t\|^2 + \gamma^2(\beta - 2)\|\boldsymbol{g}_t\|^2 \\
&\quad + \frac{\gamma^2}{\beta}\|\hat{\boldsymbol{v}}_{i_t}^t - \boldsymbol{D}\nabla f(\boldsymbol{x}^*)\|_{(i_t)}^2 - 2\gamma\langle \hat{\boldsymbol{v}}_{i_t}^t - \boldsymbol{D}\nabla f(\boldsymbol{x}^*), \hat{\boldsymbol{x}}_t - \boldsymbol{x}^* \rangle_{(i_t)} \\
&= \|\boldsymbol{x}_t - \boldsymbol{x}^*\|^2 + 2\gamma\langle \boldsymbol{g}_t, \hat{\boldsymbol{x}}_t - \boldsymbol{x}_t \rangle + \gamma^2(\beta - 1)\|\boldsymbol{g}_t\|^2 \\
&\quad + \frac{\gamma^2}{\beta}\|\hat{\boldsymbol{v}}_{i_t}^t - \boldsymbol{D}_{i_t}\nabla f(\boldsymbol{x}^*)\|^2 - 2\gamma\langle \hat{\boldsymbol{v}}_{i_t}^t - \boldsymbol{D}_{i_t}\nabla f(\boldsymbol{x}^*), \hat{\boldsymbol{x}}_t - \boldsymbol{x}^* \rangle. \qquad (77)
\end{aligned}
$$
$$
\text{(as } [\hat{\boldsymbol{v}}_{i_t}^t]_{T_{i_t}} = \hat{\boldsymbol{v}}_{i_t}^t)
$$

We now use the property that $i_t$ is independent of $\hat{\boldsymbol{x}}_t$ (which we enforce by reading $\hat{\boldsymbol{x}}_t$ before picking $i_t$, see Section 3), together with the unbiasedness of the gradient update $\hat{\boldsymbol{v}}_{i_t}^t$ ($\mathbf{E}\hat{\boldsymbol{v}}_{i_t}^t = \nabla f(\hat{\boldsymbol{x}}_t)$) and the definition of $\boldsymbol{D}$ to simplify the following expression as follows:

$$
\begin{aligned}
\mathbf{E}\langle \hat{\boldsymbol{v}}_{i_t}^t - \boldsymbol{D}_{i_t}\nabla f(\boldsymbol{x}^*), \hat{\boldsymbol{x}}_t - \boldsymbol{x}^* \rangle &= \langle \nabla f(\hat{\boldsymbol{x}}_t) - \nabla f(\boldsymbol{x}^*), \hat{\boldsymbol{x}}_t - \boldsymbol{x}^* \rangle \\
&\geq \frac{\mu}{2}\|\hat{\boldsymbol{x}}_t - \boldsymbol{x}^*\|^2 + B_f(\hat{\boldsymbol{x}}_t, \boldsymbol{x}^*), \qquad (78)
\end{aligned}
$$

where the last inequality comes from Lemma 1. Taking conditional expectations on (77) we get:

$$
\begin{aligned}
\mathbf{E}\|\boldsymbol{x}_{t+1} - \boldsymbol{x}^*\|^2 &\leq \|\boldsymbol{x}_t - \boldsymbol{x}^*\|^2 + 2\gamma\mathbf{E}\langle \boldsymbol{g}_t, \hat{\boldsymbol{x}}_t - \boldsymbol{x}_t \rangle + \gamma^2(\beta - 1)\mathbf{E}\|\boldsymbol{g}_t\|^2 \qquad (79) \\
&\quad + \frac{\gamma^2}{\beta}\mathbf{E}\|\hat{\boldsymbol{v}}_{i_t}^t - \boldsymbol{D}_{i_t}\nabla f(\boldsymbol{x}^*)\|^2 - \gamma\mu\|\hat{\boldsymbol{x}}_t - \boldsymbol{x}^*\|^2 - 2\gamma B_f(\hat{\boldsymbol{x}}_t, \boldsymbol{x}^*) \\
&\leq (1 - \frac{\gamma\mu}{2})\|\boldsymbol{x}_t - \boldsymbol{x}^*\|^2 + 2\gamma\mathbf{E}\langle \boldsymbol{g}_t, \hat{\boldsymbol{x}}_t - \boldsymbol{x}_t \rangle + \gamma^2(\beta - 1)\mathbf{E}\|\boldsymbol{g}_t\|^2 \\
&\quad + \frac{\gamma^2}{\beta}\mathbf{E}\|\hat{\boldsymbol{v}}_{i_t}^t - \boldsymbol{D}_{i_t}\nabla f(\boldsymbol{x}^*)\|^2 + \gamma\mu\|\hat{\boldsymbol{x}}_t - \boldsymbol{x}_t\|^2 - 2\gamma B_f(\hat{\boldsymbol{x}}_t, \boldsymbol{x}^*) \\
&\qquad \text{(using } \|a + b\|^2 \leq 2\|a\|^2 + 2\|b\|^2 \text{ on } \|\boldsymbol{x}_t - \hat{\boldsymbol{x}}_t + \hat{\boldsymbol{x}}_t - \boldsymbol{x}^*\|^2) \\
&\leq (1 - \frac{\gamma\mu}{2})\|\boldsymbol{x}_t - \boldsymbol{x}^*\|^2 + \gamma^2(\beta - 1)\mathbf{E}\|\boldsymbol{g}_t\|^2 + \gamma\mu\|\hat{\boldsymbol{x}}_t - \boldsymbol{x}_t\|^2 + 2\gamma\mathbf{E}\langle \boldsymbol{g}_t, \hat{\boldsymbol{x}}_t - \boldsymbol{x}_t \rangle \\
&\quad - 2\gamma B_f(\hat{\boldsymbol{x}}_t, \boldsymbol{x}^*) + \frac{4\gamma^2 L}{\beta}B_f(\hat{\boldsymbol{x}}_t, \boldsymbol{x}^*) + \frac{2\gamma^2}{\beta}\mathbf{E}\|\hat{\boldsymbol{\alpha}}_{i_t}^t - \nabla f_{i_t}(\boldsymbol{x}^*)\|^2. \qquad (80)
\end{aligned}
$$
(using Lemma 8 on the variance terms)

Since we also have:

$$
\hat{\boldsymbol{x}}_t - \boldsymbol{x}_t = \gamma \sum_{u=(t-\tau)_+}^{t-1} \boldsymbol{G}_u^t \boldsymbol{g}(\hat{\boldsymbol{x}}_u, \hat{\boldsymbol{\alpha}}^u, i_u), \qquad (81)
$$

the effect of asynchrony for the perturbed iterate updates was already derived in a very similar setup in Leblond et al. (2017). We re-use the following bounds from their Appendix C.4:[7]

$$
\mathbb{E}\|\hat{\boldsymbol{x}}_t - \boldsymbol{x}_t\|^2 \leq \gamma^2(1 + \sqrt{\Delta}\tau) \sum_{u=(t-\tau)_+}^{t-1} \mathbb{E}\|\boldsymbol{g}_u\|^2, \qquad \text{Leblond et al. (2017, Eq. (48))}
$$
$$(82)$$

$$
\mathbb{E}\langle \boldsymbol{g}_t, \hat{\boldsymbol{x}}_t - \boldsymbol{x}_t \rangle \leq \frac{\gamma\sqrt{\Delta}}{2} \sum_{u=(t-\tau)_+}^{t-1} \mathbb{E}\|\boldsymbol{g}_u\|^2 + \frac{\gamma\sqrt{\Delta}\tau}{2}\mathbb{E}\|\boldsymbol{g}_t\|^2. \qquad \text{Leblond et al. (2017, Eq. (46))}.
$$
$$(83)$$

Because the updates on $\boldsymbol{\alpha}$ are the same for PROXASAGA as for ASAGA, we can re-use the same argument arising in the proof of Leblond et al. (2017, Lemma 2) to get the following bound on $\mathbb{E}\|\hat{\boldsymbol{\alpha}}_{i_t}^t - \nabla f_{i_t}(\boldsymbol{x}^*)\|^2$:

$$\mathbb{E}\|\hat{\boldsymbol{\alpha}}_{i_t}^t - \nabla f_{i_t}(\boldsymbol{x}^*)\|^2 \le \underbrace{\frac{2L}{n}\sum_{u=1}^{t-1}(1-\frac{1}{n})^{(t-2\tau-u-1)_+}\mathbb{E}B_f(\hat{\boldsymbol{x}}_u,\boldsymbol{x}^*)+2L(1-\frac{1}{n})^{(t-\tau)_+}\tilde{e}_0}_{\text{Henceforth denoted } H_t}, \quad (84)$$

where $\tilde{e}_0 := \frac{1}{2L}\mathbb{E}\|\boldsymbol{\alpha}_i^0 - f_i'(\boldsymbol{x}^*)\|^2$. This bound is obtained by analyzing which gradient could be the source of $\boldsymbol{\alpha}_{i_t}$ in the past (taking in consideration the inconsistent writes), and then applying Lemma 2 on the $\mathbb{E}\|\nabla f(\hat{\boldsymbol{x}}_u) - \nabla f(\boldsymbol{x}^*)\|^2$ terms, explaining the presence of $B_f(\hat{\boldsymbol{x}}_u,\boldsymbol{x}^*)$ terms.[8] The inequality (84) corresponds to Eq. (56) and (57) in Leblond et al. (2017).

By taking the full expectation of (80) and plugging the above inequalities back, we obtain an inequality similar to Leblond et al. (2017, Master inequality (28)) which describes how the error terms $a_t := \mathbb{E}\|\boldsymbol{x}_t - \boldsymbol{x}^*\|^2$ of the virtual iterates are related:

$$\begin{aligned} a_{t+1} \le &(1-\frac{\gamma\mu}{2})a_t + \frac{4\gamma^2 L}{\beta}(1-\frac{1}{n})^{(t-\tau)_+}\tilde{e}_0 \\ &+ \gamma^2\Big[\beta-1+\sqrt{\Delta}\tau\Big]\mathbb{E}\|\boldsymbol{g}_t\|^2 + \Big[\gamma^2\sqrt{\Delta}+\gamma^3\mu(1+\sqrt{\Delta}\tau)\Big]\sum_{u=(t-\tau)_+}^{t}\mathbb{E}\|\boldsymbol{g}_u\|^2 \quad (85)\\ &- 2\gamma\mathbb{E}B_f(\hat{\boldsymbol{x}}_t,\boldsymbol{x}^*) + \frac{4\gamma^2 L}{\beta}\mathbb{E}B_f(\hat{\boldsymbol{x}}_t,\boldsymbol{x}^*) + \frac{4\gamma^2 L}{\beta n}H_t. \end{aligned}$$

We now have a promising inequality with a contractive term and several quantities that we need to bound. In order to achieve our final result, we introduce the same Lyapunov function as in Leblond et al. (2017):

$$\mathcal{L}_t := \sum_{u=0}^{t}(1-\rho)^{t-u}a_u,$$

where $\rho$ is a target rate factor for which we will provide a value later on. Proving that this Lyapunov function is bounded by a contraction will finish our proof. We have:

$$\begin{aligned} \mathcal{L}_{t+1} = \sum_{u=0}^{t+1}(1-\rho)^{t+1-u}a_u &= (1-\rho)^{t+1}a_0 + \sum_{u=1}^{t+1}(1-\rho)^{t+1-u}a_u \\ &= (1-\rho)^{t+1}a_0 + \sum_{u=0}^{t}(1-\rho)^{t-u}a_{u+1}. \quad (86) \end{aligned}$$

We now plug our new bound on $a_{t+1}$, (85):

$$\begin{aligned} \mathcal{L}_{t+1} \le (1-\rho)^{t+1}a_0 + \sum_{u=0}^{t}(1-\rho)^{t-u}\Big[&(1-\frac{\gamma\mu}{2})a_u + \frac{4\gamma^2 L}{\beta}(1-\frac{1}{n})^{(u-\tau)_+}\tilde{e}_0 \\ &+ \gamma^2(\beta-1+\sqrt{\Delta}\tau)\mathbb{E}\|\boldsymbol{g}_u\|^2 \\ &+ (\gamma^2\sqrt{\Delta}+\gamma^3\mu(1+\sqrt{\Delta}\tau))\sum_{v=(u-\tau)_+}^{u}\mathbb{E}\|\boldsymbol{g}_v\|^2 \quad (87)\\ &- 2\gamma\mathbb{E}B_f(\hat{\boldsymbol{x}}_u,\boldsymbol{x}^*) + \frac{4\gamma^2 L}{\beta}\mathbb{E}B_f(\hat{\boldsymbol{x}}_u,\boldsymbol{x}^*) + \frac{4\gamma^2 L}{\beta n}H_u\Big]. \end{aligned}$$

After regrouping similar terms, we get:

$$\mathcal{L}_{t+1} \le (1-\rho)^{t+1}(a_0+A\tilde{e}_0) + (1-\frac{\gamma\mu}{2})\mathcal{L}_t + \sum_{u=0}^{t}s_u^t\mathbb{E}\|\boldsymbol{g}_u\|^2 + \sum_{u=1}^{t}r_u^t\mathbb{E}B_f(\hat{\boldsymbol{x}}_u,\boldsymbol{x}^*). \quad (88)$$

Now, provided that we can prove that under certain conditions the $s_u^t$ and $r_u^t$ terms are all negative (and that the $A$ term is not too big), we can drop them from the right-hand side of (88) which will allow us to finish the proof.

Let us compute these terms. Let $q := \frac{1 - 1/n}{1 - \rho}$ and we assume in the rest that $\rho < 1/n$.

**Computing $A$.** We have:

$$\frac{4\gamma^2 L}{\beta} \sum_{u=0}^{t} (1 - \rho)^{t-u} (1 - \frac{1}{n})^{(u-\tau)_+} \leq \frac{4\gamma^2 L}{\beta} (1 - \rho)^t (1 - \rho)^{-\tau} (\tau + 1 + \frac{1}{1 - q}) \tag{89}$$

$$\text{from Leblond et al. (2017, Eq (75))}$$

$$= (1 - \rho)^{t+1} \underbrace{\frac{4\gamma^2 L}{\beta} (1 - \rho)^{-\tau-1} (\tau + 1 + \frac{1}{1 - q})}_{:=A} . \tag{90}$$

**Computing $s_u^t$.** Since we have:

$$\sum_{u=0}^{t} (1 - \rho)^{t-u} \sum_{v=(u-\tau)_+}^{u-1} \mathbb{E}\|\boldsymbol{g}_u\|^2 \leq \tau(1 - \rho)^{-\tau} \sum_{u=0}^{t} (1 - \rho)^{t-u} \mathbb{E}\|\boldsymbol{g}_u\|^2 , \tag{91}$$

we have for all $0 \leq u \leq t$:

$$s_u^t \leq (1 - \rho)^{t-u} \left[ \gamma^2 (\beta - 1 + \sqrt{\Delta}\tau) + \tau(1 - \rho)^{-\tau} (\gamma^2 \sqrt{\Delta} + \gamma^3 \mu(1 + \sqrt{\Delta}\tau)) \right] . \tag{92}$$

**Computing $r_u^t$.** To analyze these quantities, we need to compute: $\sum_{u=0}^{t} (1 - \rho)^{t-u} \sum_{v=1}^{u-1} (1 - \frac{1}{n})^{(u-2\tau-v-1)_+}$. Fortunately, this is already done in Leblond et al. (2017, Eq (66)), and thus we know that for all $1 \leq u \leq t$:

$$r_u^t \leq (1 - \rho)^{t-u} \left[ -2\gamma + \frac{4\gamma^2 L}{\beta} + \frac{4L\gamma^2}{n\beta} (1 - \rho)^{-2\tau-1} \left( 2\tau + \frac{1}{1 - q} \right) \right] , \tag{93}$$

recalling that $q := \frac{1 - 1/n}{1 - \rho}$ and that we assumed $\rho < \frac{1}{n}$.

We now need some assumptions to further analyze these quantities. We make simple choices for simplicity, though a tighter analysis is possible. To get manageable (and simple) constants, we follow Leblond et al. (2017, Eq. (82) and (83)) and assume:

$$\rho \leq \frac{1}{4n}; \quad \tau \leq \frac{n}{10} . \tag{94}$$

This tells us:

$$\frac{1}{1 - q} \leq \frac{4n}{3}$$

$$(1 - \rho)^{-k\tau-1} \leq \frac{4}{3} \qquad \text{for } 0 \leq k \leq 2 . \qquad \text{(using Bernouilli's inequality)}$$

Additionally, we set $\beta = \frac{1}{2}$. Equation (92) thus becomes:

$$s_u^t \leq \gamma^2 (1 - \rho)^{t-u} \left[ -\frac{1}{2} + \sqrt{\Delta}\tau + \frac{4}{3} (\sqrt{\Delta}\tau + \gamma\mu\tau(1 + \sqrt{\Delta}\tau)) \right] . \tag{95}$$

We see that for $s_u^t$ to be negative, we need $\tau = \mathcal{O}(\frac{1}{\sqrt{\Delta}})$. Let us assume that $\tau \leq \frac{1}{10\sqrt{\Delta}}$. We then get:

$$s_u^t \leq \gamma^2 (1 - \rho)^{t-u} \left[ -\frac{1}{2} + \frac{1}{10} + \frac{4}{30} + \gamma\mu\tau \frac{4}{3} \frac{11}{10} \right] . \tag{96}$$

Thus, the condition under which all $s_u^t$ are negative boils down to:

$$\gamma\mu\tau \leq \frac{2}{11} . \tag{97}$$

Now looking at the $r_u^t$ terms given our assumptions, the inequality (93) becomes:

$$
\begin{aligned}
r_u^t &\leq (1-\rho)^{t-u}\left[-2\gamma + 8\gamma^2 L + \frac{8\gamma^2 L}{n}\frac{4}{3}\left(\frac{n}{5} + \frac{4n}{3}\right)\right] \\
&\leq (1-\rho)^{t-u}\left(-2\gamma + 36\gamma^2 L\right).
\end{aligned}
\tag{98}
$$

The condition for all $r_u^t$ to be negative then can be simplified down to:

$$
\gamma \leq \frac{1}{18L}.
\tag{99}
$$

We now have a promising inequality for proving that our Lyapunov function is bounded by a contraction. However we have defined $\mathcal{L}_t$ in terms of the virtual iterate $x_t$, which means that our result would only hold for a given $T$ fixed in advance, as is the case in Mania et al. (2017). Fortunately, we can use the same trick as in Leblond et al. (2017, Eq. (97)): we simply add $\gamma B_f(\hat{x}_t, x^*)$ to both sides in (88). $r_t^t$ is replaced by $r_t^t + \gamma$, which makes for a slightly worse bound on $\gamma$ to ensure linear convergence:

$$
\gamma \leq \frac{1}{36L}.
\tag{100}
$$

For this small cost, we get a contraction bound on $B_f(\hat{x}_t, x^*)$, and thus by the strong convexity of $f$ (see (9)) we get a contraction bound for $\mathbb{E}\|\hat{x}_t - x^*\|^2$.

**Recap.** Let us use $\rho = \frac{1}{4n}$ and $\gamma := \frac{a}{L}$. Then the conditions (97) and (100) on the step size $\gamma$ reduce to:

$$
a \leq \frac{1}{36}\min\{1, \frac{72}{11}\frac{\kappa}{\tau}\}.
\tag{101}
$$

Moreover, the condition:

$$
\tau \leq \frac{1}{10\sqrt{\Delta}}
\tag{102}
$$

is sufficient to also ensure that (94) is satisfied as $\Delta \in [\frac{1}{n}, 1]$, and thus $\frac{1}{\sqrt{\Delta}} \leq \sqrt{n} \leq n$.

Thus under the conditions (101) and (102), we have that all $s_u^t$ and $r_u^t$ terms are negative and we can rewrite the recurrent step of our Lyapunov function as:

$$
\mathcal{L}_{t+1} \leq \gamma\mathbb{E}B_f(\hat{x}_t) + \mathcal{L}_{t+1} \leq (1-\rho)^{t+1}(a_0 + A\tilde{e}_0) + (1 - \frac{\gamma\mu}{2})\mathcal{L}_t.
\tag{103}
$$

By unrolling the recursion (103), we can carefully combine the effect of the geometric term $(1-\rho)$ with the one of $(1 - \frac{\gamma\mu}{2})$. This was already done in Leblond et al. (2017, Apx C.9, Eq. (101) to (103)), with a trick to handle various boundary cases, yielding the overall rate:

$$
\mathbb{E}B_f(\hat{x}_t, x^*) \leq (1-\rho^*)^{t+1}\hat{C}_0,
\tag{104}
$$

where $\rho^* = \min\{\frac{1}{5n}, a\frac{2}{5\kappa}\}$ (that we simplified to $\rho^* = \frac{1}{5}\min\{\frac{1}{n}, a\frac{1}{\kappa}\}$ in the theorem statement). To get the final constant, we need to bound $A$. We have:

$$
\begin{aligned}
A &= \frac{4\gamma^2 L}{\beta}(1-\rho)^{-\tau-1}(\tau + 1 + \frac{1}{1-q}) \\
&\leq 8\gamma^2 L\frac{4}{3}(\frac{n}{10} + 1 + \frac{4n}{3}) \\
&\leq 26\gamma^2 Ln \\
&\leq \gamma n.
\end{aligned}
\tag{105}
$$

This is the same bound on $A$ that was used by Leblond et al. (2017) and so we obtain the same constant as their Eq. (104):

$$
\hat{C}_0 := \frac{21n}{\gamma}(\|x_0 - x^*\|^2 + \gamma\frac{n}{2L}\mathbb{E}\|\alpha_i^0 - \nabla f_i(x^*)\|^2).
\tag{106}
$$

Note that $\hat{C}_0 = \mathcal{O}(\frac{n}{\gamma}C_0)$ with $C_0$ defined as in Theorem **??**.

Now, using the strong convexity of $f$ via (9), we get:

$$\mathbb{E}\|\hat{\boldsymbol{x}}_t - \boldsymbol{x}^*\|^2 \leq \frac{2}{\mu}\mathbb{E}B_f(\hat{\boldsymbol{x}}_t, \boldsymbol{x}^*) \leq (1 - \rho^*)^{t+1}\tilde{C}_0, \tag{107}$$

where $\tilde{C}_0 = \mathcal{O}(\frac{n\kappa}{a}C_0)$.

This finishes the proof for Theorem 2. $\qquad\square$

**Corollary 3** (Speedup). *Suppose* $\tau \leq \frac{1}{10\sqrt{\Delta}}$. *If* $\kappa \geq n$, *then using the step size* $\gamma = 1/36L$, PROXAS-AGA *converges geometrically with rate factor* $\Omega(\frac{1}{\kappa})$. *If* $\kappa < n$, *then using the step size* $\gamma = 1/36n\mu$, PROXASAGA *converges geometrically with rate factor* $\Omega(\frac{1}{n})$. *In both cases, the convergence rate is the same as Sparse Proximal* SAGA *and* PROXASAGA *is thus linearly faster than its sequential counterpart up to a constant factor. Note that in both cases* the step size does not depend on $\tau$.

*Furthermore, if* $\tau \leq 6\kappa$, *we can use a universal step size of* $\Theta(1/L)$ *to get a similar rate for* PROX-ASAGA *than Sparse Proximal* SAGA, *thus making it adaptive to local strong convexity since the knowledge of* $\kappa$ *is not required.*

*Proof.* If $\kappa \geq n$, the rate factor of Sparse Proximal SAGA is $1/\kappa$. To get the same rate factor, we need to choose $a = \Omega(1)$, which we can fortunately do since $\kappa \geq n \geq \sqrt{n} \geq 10\frac{1}{10\sqrt{\Delta}} \geq 10\tau$.

If $\kappa < n$, then the rate factor of Sparse Proximal SAGA is $1/n$. Any choice of $a$ bigger than $\Omega(\kappa/n)$ gives us the same rate factor for PROXASAGA. Since $\tau \leq \sqrt{n}/10$ we can pick such an $a$ without violating the condition of Theorem 2. $\qquad\square$

## Appendix D    Comparison with related work

In this section, we relate our theoretical results and proof technique with the related literature.

**Speedups.** Our speedup regimes are comparable with the best ones obtained in the smooth case, including Niu et al. (2011); Reddi et al. (2015), even though unlike these papers, we support inconsistent reads and nonsmooth objective functions. The one exception is Leblond et al. (2017), where the authors prove that their algorithm, ASAGA, can obtain a linear speedup even without sparsity in the well-conditioned regime. In contrast, PROXASAGA always requires some sparsity. Whether this property for smooth objective functions could be extended to the composite case remains an open problem.

**Coordinate Descent.** We compare our approach for composite objective functions to its most natural competitor: ASYSPCD (Liu & Wright, 2015), an asynchronous stochastic coordinate descent algorithm. While ASYSPCD also exhibits linear speedups, subject to a condition on $\tau$, one has to be especially careful when trying to compare these conditions.

First, while in theory the iterations of both algorithms have the same cost, in practice various tricks are introduced to save on computation, yielding different costs per updates.[9] Second, the bound on $\tau$ for the coordinate descent algorithm depends on $p$, the dimensionality of the problem, whereas ours involves $n$, the number of data points. Third, a more subtle issue is that $\tau$ is not affected by the same quantities for both algorithms.[10] See Appendix D.1 for a more detailed explanation of the differences between the bounds.

In the best case scenario (where the components of the gradient are uncorrelated, a somewhat unrealistic setting), ASYSPCD can get a near-linear speedup for $\tau$ as big as $\sqrt[4]{p}$. Our result states that $\tau = \mathcal{O}(1/\sqrt{\Delta})$ is necessary for a linear speedup. This means in case $\Delta \leq 1/\sqrt{p}$ our bound is better than the one obtained for ASYSPCD. Recalling that $1/n \leq \Delta \leq 1$, it appears that PROXASAGA is favored when $n$ is bigger than $\sqrt{p}$ whereas ASYSPCD may have a better bound otherwise, though this comparison should be taken with a grain of salt given the assumptions we had to make to arrive at comparable quantities.

Furthermore, one has to note that while Liu & Wright (2015) use the classical labeling scheme inherited from Niu et al. (2011), they still assume in their proof that the $i_t$ are uniformly distributed and that their gradient estimators are conditionally unbiased – though neither property is verified in the general asynchronous setting. Finally, we note that ASYSPCD (as well as its incremental variant Async-PROXSVRCD) assumes that the computation and assignment of the proximal operator is an atomic step, while we do not make such assumption.

**SVRG.** The Async-ProxSVRG algorithm of Meng et al. (2017) also exhibits theoretical linear speedups subject to the same condition as ours. However, the analyzed algorithm uses dense updates and consistent read and writes. Although they make the analysis easier, these two factors introduce costly bottlenecks and prevent linear speedups in running time. Furthermore, here again the classical labeling scheme is used together with the unverified conditional unbiasedness condition.

**Doubly stochastic algorithms.** The Async-PROXSVRCD algorithm from Meng et al. (2017); Gu et al. (2016) has a maximum allowable stepsize[11] that is in $\mathcal{O}(1/pL)$, whereas the maximum step size for PROXASAGA is in $\Omega(1/L)$, so can be up to $p$ times bigger. Consequently, PROXASAGA enjoys much faster theoretical convergence rates. Unfortunately, we could not find a condition for linear speedups to compare to. We also note that their algorithm is not appropriate in a sparse features setting. This is illustrated in an empirical comparison in Appendix F where we see that

their convergence in number of iterations is orders of magnitude slower than appropriate algorithms like SAGA or PROXASAGA.

### Appendix D.1 Comparison of bounds with Liu & Wright (2015)

**Iteration costs.** For both PROXASAGA and ASYSPCD, the average cost of an iteration is $\mathcal{O}(n\overline{S})$ (where $\overline{S}$ is the average support size). In the case of PROXASAGA (see Algorithm 1), at each iteration the most costly operation is the computation of $\overline{\alpha}$, while in the general case we need to compute a full gradient for ASYSPCD.

In order to reduce these prohibitive computation costs, several tricks are introduced. Although they lead to much improved empirical performance, it should be noted that in both cases these tricks are not covered by the theory. In particular, the unbiasedness condition can be violated.

In the case of PROXASAGA, we store the average gradient term $\overline{\alpha}$ in shared memory. The cost of each iteration then becomes the size of the extended support of the partial gradient selected at random at this iteration, hence it is in $\mathcal{O}(\Delta_l)$, where $\Delta_l := \max_{i=1..n} |T_i|$.

For ASYSPCD, following Peng et al. (2016) we can store intermediary quantities for specific losses (e.g. $\ell_1$-regularized logistic regression). The cost of an iteration then becomes the number of data points whose extended support includes the coordinate selected at random at this iteration, hence it is in $\mathcal{O}(n\Delta)$.

The relative difference in update cost of both algorithms then depends heavily on the data matrix: if the partial gradients usually have a extended support but coordinates belong to few of them (this can be the case if $n \ll p$ for example), then the iterations of ASYSPCD can be cheaper than those of PROXASAGA. Conversely, if data points usually have small extended support but coordinates belong to many of them (which can happen when $p \ll n$ for example), then the updates of PROXASAGA are the cheaper ones.

**Dependency of $\tau$ on the data matrix.** In the case of PROXASAGA the sizes of the extended support of each data point are important – they are directly linked to the cost of each iteration. Identical iteration costs for each data point do not influence $\tau$, whereas heterogeneous costs may cause $\tau$ to increase substantially. In contrast, in the case of ASYSPCD, the relevant parts of the data matrix are the number of data points each dimension touches – for much the same reason. In the bipartite graph between data points and dimensions, either the left or the right degrees matter for $\tau$, depending on which algorithm you choose.

In order to compare their respective bounds, we have to make the assumption that the iteration costs are homogeneous, which means that each data point has the same support size and each dimension is active in the same number of data points. This implies that $\tau$ is the same quantity for both algorithms.

**Best case scenario bound for AsySPCD.** The result obtained in Liu & Wright (2015) states that if $\tau^2 \Lambda = \mathcal{O}(\sqrt{p})$, ASYSPCD can get a near-linear speedup (where $\Lambda$ is a measure of the interactions between the components of the gradient, with $1 \leq \Lambda \leq \sqrt{p}$). In the best possible scenario where $\Lambda = 1$ (which means that the coordinates of the gradients are completely uncorrelated), $\tau$ can be as big as $\sqrt[4]{p}$.

## Appendix E   Implementation details

**Initialization.**   In the Sparse Proximal SAGA algorithm and its asynchronous variant, PROXAS-AGA, the vector $\boldsymbol{x}$ can be initialized arbitrarily. The memory terms $\boldsymbol{\alpha}_i$ can be initialized to any vector that verifies $\text{supp}(\boldsymbol{\alpha}_i) = \text{supp}(\nabla f_i)$. In practice we found that the initialization $\boldsymbol{\alpha}_i = \mathbf{0}$ is very fast to set up and often outperforms more costly initializations.

With this initialization, the gradient approximation before the first update of the memory terms becomes $\nabla f_i(\boldsymbol{x}) + \boldsymbol{D}_i \overline{\boldsymbol{\alpha}}$. Since most of the values in $\boldsymbol{\alpha}$ are zero, $\overline{\boldsymbol{\alpha}}$ will tend to be small compared to $\nabla f_i(\boldsymbol{x})$, and so the gradient estimate is very close to the SGD estimate $\nabla f_i(\boldsymbol{x})$. The SGD approximation is known to have a very fast initial convergence (which, in light of Figure 1, our method inherits) and has even been used as a heuristic to use during the first epoch of variance reduced methods (Schmidt et al., 2016).

The initialization of coefficients $\boldsymbol{x}_0$ was always set to zero.

**Exact regularization.**   Computing the gradient of a smooth regularization such as the squared $\ell_2$ penalty of Eq. (6) is independent of $n$ and so we can use the exact regularizer in the update of the coefficients instead of storing it in $\boldsymbol{\alpha}$, which would also destroy the compressed storage of the memory terms described below. In practice we use this "exact regularization", multiplied by $\boldsymbol{D}_i$ to preserve the sparsity pattern.

Assuming a squared $\ell_2$ regularization term of the form $\frac{\lambda}{2}$, the gradient estimate in (SPS) becomes (note the extra $\lambda \boldsymbol{x}$)

$$\boldsymbol{v}_i = \nabla f_i(\boldsymbol{x}) - \boldsymbol{\alpha}_i + \boldsymbol{D}_i(\overline{\boldsymbol{\alpha}} + \lambda \boldsymbol{x}) \ . \tag{108}$$

**Storage of memory terms.**   The storage requirements for this method is in the worst case a table of size $n \times p$. However, as for SAG and SAGA, for linearly parametrized loss functions of the form $f_i(\boldsymbol{x}) = \ell(\boldsymbol{a}_i^T \boldsymbol{x})$, where $\ell$ is some real-valued function and $(\boldsymbol{a}_i)_{i=1}^n$ are samples associated with the learning problem, this can be reduced to a table of size $n$ (Schmidt et al., 2016, §4.1). This includes popular linear models such as least squares or logistic regression with $\ell$ the squared or logistic function, respectively.

The reduce storage comes from the fact that in this case the partial gradients have the structure

$$\nabla f_i(\boldsymbol{x}) = \boldsymbol{a}_i \underbrace{\ell'(\boldsymbol{a}_i^T \boldsymbol{x})}_{\text{scalar}} \ . \tag{109}$$

Since $\boldsymbol{a}_i$ is independent of $\boldsymbol{x}$, we only need to store the scalar $\ell'(\boldsymbol{a}_i^T \boldsymbol{x})$. This decomposition also explains why $\nabla f_i$ inherits the sparsity pattern of $\boldsymbol{a}_i$.

**Atomic updates.**   Most modern processors have support for atomic operations with minimal overhead. In our case, we implemented a double-precision atomic type using the C++11 atomic features (`std::atomic<double>`). This type implements atomic operations through the compare and swap semantics.

Empirically, we have found it necessary to implement atomic operations at least in the vector $\boldsymbol{\alpha}$ and $\overline{\boldsymbol{\alpha}}$ to reach arbitrary precision. If non-atomic operations are used, the method converges only to a limited precision (around normalized function suboptimality of $10^{-3}$), which might be sufficient for some machine learning applications but which we found not satisfying from an optimization point of view.

**AsySPCD.**   Following (Peng et al., 2016) we keep the vector $(\boldsymbol{a}_i^T \boldsymbol{x})_{i=1}^n$ in memory and update it at each iteration using atomic updates.

**Hardware and software.**   All experiments were run on a Dell PowerEdge 920 machine with 4 Intel Xeon E7-4830v2 processors with 10 2.2GHz cores each and 384GB 1600 Mhz RAM. The PROX-ASAGAand ASYSPCD code was implemented on C++ and binded in Python. The FISTA code is implemented in pure Python using NumPY and SciPy for matrix computations (in this case the bottleneck is in large sparse matrix-vector operations for which efficient BLAS routines were used). Our PROXASAGA implementation can be downloaded from http://github.com/fabianp/ProxASAGA.

# Appendix F    Experiments

All datasets used for the experiments were downloaded from the LibSVM dataset suite.[12]

## Appendix F.1    Comparison of ProxASAGA with other sequential methods

We provide a comparison between the Sparse Proximal SAGA and related methods in the sequential case. We compare against two methods: the MRBCD method of Zhao et al. (2014) (which forms the basis of Async-PROXSVRCD) and the vanilla implementation of SAGA (Defazio et al., 2014), which does not have the ability to perform sparse updates. We compare in terms of both passes through the data (epochs) and time. We use the same step size for all methods $(1/3L)$. Due to the slow convergence of some methods, we use a smaller dataset than the ones used in §4. Dataset RCV1 has $n = 697, 641, d = 47, 236$ and a density of $0.15$, while Covtype is a dense dataset with $n = 581, 012, d = 54$.

Figure 2: Suboptimality of different sequential algorithms. Each marker represents one pass through the dataset.

We observe that for the convergence behavior in terms of number of passes, Sparse Proximal SAGA performs as well as vanilla SAGA, though the latter requires dense updates at every iteration (Fig. 2 top left). On the other hand, in terms of running time, our implementation of Sparse Proximal SAGA is much more efficient than the other methods for sparse input (Fig. 2 top right). In the case of dense input (Fig. 2 bottom), the three methods perform similarly.

**A note on the performance of MRBCD.**    It may appear surprising that Sparse Proximal SAGA outperforms MRBCD so dramatically on sparse datasets. However, one should note that MRBCD is a doubly stochastic algorithm where both a random data point and a random coordinate are sampled for each iteration. If the data matrix is very sparse, then the probability that the sampled coordinate is in the support of the sampled data point becomes very low. This means that the gradient estimator term only contains the reference gradient term of SVRG, which only changes once per epoch. As a result, this estimator becomes very coarse and produces a slower empirical convergence.

This is reflected in the theoretical results given in Zhao et al. (2014), where the epoch size needed to get linear convergence are $k$ times bigger than the ones required by plain SVRG, where $k$ is the size of the set of blocks of coordinates.

## Appendix F.2   Theoretical speedups.

In the experimental section, we have shown experimental speedup results where suboptimality was a function of the running time. This measure encompasses both theoretical algorithmic optimization properties and hardware overheads (such as contention of shared memory) which are not taken into account in our analysis.

In order to isolate these two effects, we now plot our speedup results in Figure 3 where suboptimality is a function of the number of iterations; thus, we abstract away any potential hardware overhead. To do so, we implement a global counter which is sparsely updated (every 100 iterations for example) in order not to modify the asynchrony of the system. This counter is used only for plotting purposes and is not needed otherwise. Specifically, we define the theoretical speedup as:

$$\text{theoretical speedup} := (\text{number of cores}) \frac{\text{number of iterations for sequential algorithm}}{\text{total number of iterations for parallel algorithm}} .$$

Figure 3: **Theoretical optimization speedups for $\ell_1+\ell_2$-regularized logistic regression**. Speedup as measured by the number of iterations required to reach $10^{-5}$ suboptimality for PROXASAGA and ASYSPCD. In FISTA the iterates are the same with different cores and so matches the "ideal" speedup.

We see clearly that the theoretical speedups obtained by both PROXASAGAand ASYSPCD are linear (i.e. ideal). As we observe worse results in running time, this means that the hardware overheads of asynchronous methods are quite significant.

## Appendix F.3   Timing benchmarks

We now provide the time it takes for the different methods with 10 cores to reach a suboptimality of $10^{-10}$. All results are in hours.

| Dataset | PROXASAGA | ASYSPCD | FISTA |
|---|---|---|---|
| **KDD 2010** | 1.01 | 13.3 | 5.2 |
| **KDD 2012** | 0.09 | 26.6 | 8.3 |
| **Criteo** | 0.14 | 33.3 | 6.6 |

## Appendix F.4   Hyperparameters

The $\ell_1$-regularization parameter $\lambda_2$ was chosen as to give around 10% of non-zero features. The exact chosen values are the following: $\lambda_2 = 10^{-11}$ for KDD 2010, $\lambda_2 = 10^{-16}$ for KDD 2012 and $\lambda_2 = 4 \times 10^{-12}$ for Criteo.