[Reviews · NeurIPS 2017]

Reviewer 1



This paper considers solving the finite sum optimization via the SAGA framework. This work extends from Leblond et al. 2017 by considering the composite optimization, where the additional part is the non-smooth separable regularization. The key improvement lies on how to deal with the non-smooth regularization while obey the principle of sparse update. The proposed trick to split the non-smooth regularization looks interesting to me. The analysis basically follow the framework in Leblond et al. 2017, the proof for the asynchronous variant follows the idea in Mania et al. 2015. Major comments: - Authors need to explain the "inconsistent read" more clear. In particular, what is hat{x}_k and how to decide the index of k. Is it the same as Lian et al. 2016 and Liu and Wright 2015? What is the key difference? - It is unclear to me how to obtain the neat representation of the difference between hat{x}_t and x_t. - To obtain the linear convergence rate and speedup, this paper makes several implicit assumptions. Authors should explicitly indicate the assumption used in this paper. Minor comments / typos: - The definition of \Delta in line 217 was incorrect. Missing reference: The following is related to asynchronous greedy SDCA. - Asynchronous parallel greedy coordinate descent, NIPS, 2016.

Reviewer 2



This paper proposes an asynchronous variant of SAGA, a stochastic gradient method for finite sum convex optimization (Defazio et al, 2014). To my understanding, the paper addresses two challenges: asynchronicity and proximity. The authors achieve three improvements over the original SAGA. First, they propose a sparse update based on block coordinate wise using extended support for the gradients when the regularization is decomposable. Second, they design an asynchronous variant of SAGA where the delay quantity can be up to sqrt(n)/10. Finally, they can deal with nonsmooth regularizers via proximal operators. In terms of convergence results, the still achieve a linear convergence rate under the strong convexity of the overall sum function f, and individual Lipschitz gradient of fi. Although the step-size is slightly smaller than the one of SAGA, the convergence factor remains comparable. This is probably due to different assumption. In fact, the paper combines several advanced ideas from existing works such as SAGA with variance reduction, sparse updates, Hogwild, asynchorinization such as Arock, etc, to improve over all these techniques. The proof is quite clean and well organized. In my opinion, such improvements are significant and are important in practice due to obvious reasons. The numerical results also strongly support their theoretical contribution. These experiments are carried out on three large-scale data sets, and empirically show a linear speed up of the new algorithm. Overall, this paper has significant contribution both in terms of theory and experiments. It merits to be accepted for NIPS. Minor comments. Some concepts and notation should be defined. For example, support (supp), \Omega(\cdot), inconsistence read. The phrase “step size” should be consistent on all text. For example, on line 46 and line 142 they are not consistent. Line 152, delete “a” or “the”. Line 361, the y should be in bold. Line 127: remove one “a”.

Reviewer 3



The main contribution of this paper is to offer a convergence proof for minimizing sum fi(x) + g(x) where fi(x) is smooth, and g is nonsmooth, in an asynchronous setting. The problem is well-motivated; there is indeed no known proof for this, in my knowledge. A key aspect of this work is the block separability of the variable, which allows for some decomposition gain in the prox step (since the gradient of each fi may only effect certain indices of x.) This is an important practical assumption, as otherwise the prox causes locks, which may be unavoidable. (It would be interesting to see element-wise proxes, such as shrinkage, being decomposed as well.) Their are two main theoretical results. Theorem 1 gives a convergance rate for proxSAGA, which is incrementally better than a previous result. Theorem 2 gives the rate for an asynchronous setting, which is more groundbreaking. Overall I think it is a good paper, with a very nice theoretical contribution that is also practically very useful. I think it would be stronger without the sparsity assumption, but the authors also say that investigating asynchronosy without sparsity for nonsmooth functions is an open problem, so that is acknowledged. potentially major comment: - Theorem 2 relies heavily on a result from leblond 2017, which assumes smooth g's. It is not clear in the proof that the parts borrowed from Leblond 2017 does not require this assumption. minor comments about paper: - line 61, semicolon after "sparse" - line 127: "a a" minor comments about proof (mostly theorem 1): - eq. (14) is backwards (x-z, not z-x) - line 425: do you mean x+ = x - gamma vi? - Lemma 7 can divide RHS by 2 (triangle inequality in line 433 is loose) - line 435: not the third term (4th term) - line 444: I am not entirely clear why EDi = I. Di has to do with problem structure, not asynchronasy? - eq (33) those two are identical, typo somewhere with missing Ti? - line 447: Not sure where that split is coming from, and the relationship between alpha i, alpha i +, and grad fi. - line 448: there is some mixup in the second line, a nabla f_i(x*) became nabla f_i(x). Some of these seem pretty necessary to fix, but none of these seem fatal, especially since the result is very close to a previously proven result. I mostly skimmed theorem 2 proof; it looks reasonable, apart from the concern of borrowing results from Leblond (assumptions on g need to be clearer)